# Quantum Machine Learning: A Review and Case Studies

**DOI:** 10.3390/e25020287

**Published:** 2023-02-03

**Authors:** Amine Zeguendry, Zahi Jarir, Mohamed Quafafou

**Affiliations:** 1Laboratoire d’Ingénierie des Systèmes d’Information, Faculty of Sciences, Cadi Ayyad University, Marrakech 40000, Morocco; 2Laboratoire des Sciences de l’Information et des Systèmes, Unité Mixte de Recherche 7296, Aix-Marseille University, 13007 Marseille, France

**Keywords:** quantum computing, quantum algorithms, Quantum Machine Learning (QML), quantum classification, Variational Quantum Circuit (VQC), QSVM, Quanvolutional Neural Network (QNN), Variational Quantum Classifier (VQC), quantum encoding

## Abstract

Despite its undeniable success, classical machine learning remains a resource-intensive process. Practical computational efforts for training state-of-the-art models can now only be handled by high speed computer hardware. As this trend is expected to continue, it should come as no surprise that an increasing number of machine learning researchers are investigating the possible advantages of quantum computing. The scientific literature on Quantum Machine Learning is now enormous, and a review of its current state that can be comprehended without a physics background is necessary. The objective of this study is to present a review of Quantum Machine Learning from the perspective of conventional techniques. Departing from giving a research path from fundamental quantum theory through Quantum Machine Learning algorithms from a computer scientist’s perspective, we discuss a set of basic algorithms for Quantum Machine Learning, which are the fundamental components for Quantum Machine Learning algorithms. We implement the Quanvolutional Neural Networks (QNNs) on a quantum computer to recognize handwritten digits, and compare its performance to that of its classical counterpart, the Convolutional Neural Networks (CNNs). Additionally, we implement the QSVM on the breast cancer dataset and compare it to the classical SVM. Finally, we implement the Variational Quantum Classifier (VQC) and many classical classifiers on the Iris dataset to compare their accuracies.

## 1. Introduction

Machine Learning is a subset of Artificial Intelligence (AI) that aims to create models that learn from previous experience without being explicitly formulated, and it has been used extensively in several scientific and technical fields, including natural language processing, medical diagnostics, computer vision, and data mining, and so on. Many machine learning problems require the use of linear algebra to execute matrix operations since data are described as matrices. On the other hand, performing these operations on traditional computers requires a significant amount of time and computational resources. Quantum Computing is an ambitious new field that combines computer science, mathematics, and physics. This field investigates ways to use some of the special properties of quantum physics to build quantum computers that take advantage of quantum bits (qubits) that can contain 0 and 1 combinations in superposition at the same time. As a result, quantum computers can handle and process large matrices, as well as accelerate various linear algebra operations, significantly improving traditional machine learning applications. Theoretically, it should solve problems that belong to classes of complexity that traditional computers, even large giant supercomputers, will never be able to solve. Grover’s algorithm, for instance, has demonstrated a quadratic speedup for exploring unstructured databases [1], while Shor’s technique [2] illustrates that quantum computing may provide an exponential speedup in solving the traditionally difficult problem of big integer factorization. These algorithms are anticipated to include some key characteristics of quantum computation, including quantum superposition, quantum measurement, and quantum entanglement.

While machine learning is being limited by a lack of computing power, researchers are exploring the prospects of combining quantum computing and machine learning to handle classical data using machine learning algorithms. This combination of machine learning theory with the characteristics of quantum computing is a new research sub-discipline called Quantum Machine Learning (QML). Therefore, QML aims to build quantum applications for diverse machine learning algorithms, using the processing power of quantum computers and the scalability and learning capacity of machine learning algorithms.

Quantum variants of several popular algorithms for machine learning are already developed. Quantum Neural Network (QNN) described by Narayanan and Menneer [3], in which they presented the theoretical design of a QNN architecture and how the system’s components might perform relative to their traditional counterparts. Quantum Support Vector Machines (QSVM) was proposed by Rebentrost et al. [4] for solving least-squares SVM using the HHL algorithm [5] for matrix inversion in order to generate the hyperplane. In 2014, Wiebe et al. [6] provided a quantum version of k-nearest neighbors for computing the nearest neighbors relying on the Euclidean distance between the data locations, in addition to amplitude estimation which eliminates the requirement for measurement. In 2018, Dang et al. [7] also introduced an image classification model based on quantum k-nearest neighbors and parallel computing. Their model improved categorization precision and productivity. Schuld et al. [8] proposed quantum linear regression as a version of classical linear regression, and it operates in an exponential runtime with *N* dimensions of features, using quantum data, which is presented as quantum information. The quantum decision tree classifier developed by Lu et al. [9] employs quantum fidelity measurement and quantum entropy impurity.

Lloyd et al. [10] described several quantum machine learning techniques for cluster detection and cluster assignment. A quantization version of Lloyd’s method [11] is provided as a part of their k-means clustering algorithm. Furthermore, Kerenidis et al. [12] have suggested Q-means, a quantized form of the k-means method with similar results and convergence to the traditional k-means algorithm. Aïmeur et al. [13] introduced quantum K-medians clustering, which uses Grover’s search algorithm for locating the cluster’s median. In 2014, Lloyd et al. [14] developed Quantum Principal Component Analysis (QPCA), which identifies the eigenvectors relating to the unknown state’s huge eigenvalues exponentially faster than any other solution.

Inside Machine Learning, there is a more advanced field known as Reinforcement Learning. It is built on continuous learning through exploration of the environment. There exist few quantum variants of conventional Reinforcement Learning algorithms including Quantum Reinforcement Learning introduced by Dong et al. [15], which makes use of the quantum parallelism and superposition principle. They discovered that probability amplitude and quantum parallelism may aid in learning speed. For solving Dynamic Programming problems, which are deterministic forms of Markov decision problems that are studied in reinforcement learning, in [16], they have suggested various quantum algorithms as solutions. McKiernan et al. [17] developed a general algorithm to improve hybrid quantum-classical computing using reinforcement learning.

Deep Learning is a new machine learning sub-discipline. Deep Learning techniques, which demand a significant amount of storage and time consumption, are now being implemented on quantum computers. These algorithms are shown by Quantum Generative Adversarial Networks (Quantum GAN) [18,19,20] with its implementation in [21], which uses a superconducting quantum processor to learn and generate real-world handwritten digit pictures, Quantum Wasserstein Generative Adversarial Networks (Quantum WGAN) [22], which improves the scalability and stability of quantum generative adversarial model training on quantum machines, Quantum Boltzmann Machines [23,24], Quantum Autoencoders [25,26], and Quantum Convolutional Neural Networks [27,28,29], and the latter is implemented in the benchmark section.

Optimization is also important in conventional machine learning, since most of the time learning training machine learning models are based on optimizing cost functions. Numerical optimization techniques are a major field of study that tries to enhance optimization algorithm computations. Quantum-enhanced Optimization (QEO) [30], a sub-field of quantum computing, intends to augment these techniques even further, similar to conventional optimization. Quantum Approximate Optimization Algorithm and Quantum Gradient Descent [31,32] are two well-known examples of this type, which are used within Quantum Neural Networks such as Quantum Boltzman Machines [24].

In this paper, we aim to introduce quantum computing to the field of machine learning from a basics perspective all the way to its applications. The remainder of this work is structured as follows. In Section 2, we explore the background of quantum computing, the architecture of quantum computers, and an introduction to quantum algorithms. In Section 3, we start introducing several fundamental algorithms for quantum machine learning that are the fundamental components of QML algorithms and can primarily offer a performance boost over conventional machine learning algorithms, and we discuss a few popular quantum machine learning algorithms. In Section 4, we implement three machine learning algorithms in order to compare the performance of each algorithm with its classical counterparts. In the final sections, ahead of the conclusions and perspectives, we discuss some quantum computer challenges.

## 2. Background

Quantum computing uses quantum physics principles including superposition, entanglement, and quantummeasurement, to process data. Superposition is the quantum mechanical feature that permits objects to exist in several states simultaneously. Entanglement may join two or even more quantum particles in full synchronization, despite being on different sides of the universe. Quantum measurement is a process that transforms quantum information into classical information. This section provides an introduction to the basics of quantum computing, from Dirac notations through quantum algorithms.

### 2.1. Dirac (Bra-Ket) Notation

States and operators in quantum mechanics are represented as vectors and matrices, respectively. Instead of utilizing standard linear algebra symbols, Dirac notation is used to represent the vectors.

Let *a* and *b* be in $C^{2}$:Ket:
(1)$$|a\rangle =\left(\begin{array}{c} a_{1}\\ a_{2} \end{array}\right).$$

Bra:
(2)$$\langle b|={|b\rangle }^{*}={\left(\begin{array}{c} b_{1}\\ b_{2} \end{array}\right)}^{*}=\left(\begin{array}{cc} b_{1}^{*} & b_{2}^{*} \end{array}\right).$$Note that the complex conjugate of any complex number can be generated by inverting the sign of its imaginary component—for example, the complex conjugate of b=a+i×d is $b^{*}=a-i\times d$.Bra-Ket: Inner product
(3)$$\langle b|a\rangle =a_{1}b_{1}^{*}+a_{2}b_{2}^{*}={\langle a|b\rangle }^{*}.$$Ket-Bra: Outer product
(4)$$|a\rangle \langle b|=\left(\begin{array}{cc} a_{1}b_{1}^{*} & a_{1}b_{2}^{*}\\ a_{2}b_{1}^{*} & a_{2}b_{2}^{*} \end{array}\right).$$

### 2.2. Qubit

The bit is the standard measure of information in classical computing. It could exist in one of two states: 0 or 1. Quantum computers, similarly, use a “qubit” that is also called “quantum bit” [33], which represents the likelihood of an electron’s “spin up” or “spin down” when passing through some kind of a magnetic field. In traditional computing, the spin may be thought of as the value of a bit. A qubit may be considered as the complex two-dimensional Hilbert space $C^{2}$. An instantaneous qubit state can be expressed as a vector in this multidimensional Hilbert space.

An inner product between two vectors that represent qubit states in Hilbert space allows us to identify their relative positions. 〈a|b〉 represents the inner product of the vectors |a〉 and |b〉; it equals 0 if |a〉 and |b〉 are orthogonal and 1 if |a〉 = |b〉.

The states |0〉 and |1〉 can be represented with vectors, as shown below. These two are known as the computationalbasis of a two-level system. These vectors in the state space are then susceptible to matrix-based operators:(5)$$|0\rangle =\left(\begin{array}{c} 1\\ 0 \end{array}\right);|1\rangle =\left(\begin{array}{c} 0\\ 1 \end{array}\right).$$

A qubit could be in one of three states: |0〉, |1〉, or a superposition of both,
(6)$$c_{1}|0\rangle +c_{2}|1\rangle ,$$
where the coefficients $c_{1}$, $c_{2}$∈C, called amplitudes. According to the Born rule, the total of the squares of the amplitudes of all possible states in a superposition equals 1:(7)$$|c_{1}{|}^{2}+{\left|c_{2}\right|}^{2}=1.$$

When we perform a measurement, we obtain a single bit of information: 0 or 1. The most basic measurement is in the computational basis (Z-Basis: $\left\{|0\rangle ,|1\rangle \right\}$). For example, the result of measuring the state $c_{1}|0\rangle +c_{2}|1\rangle$ in the computational basis is 0 with probability $|c_{1}{|}^{2}$ and 1 with probability $|c_{2}{|}^{2}$.

There are numerous different bases; however, below are some of the most common ones:X-Basis:
$$\left\{|+\rangle :=\frac{1}{\sqrt{2}}\left(|0\rangle +|1\rangle \right),|-\rangle :=\frac{1}{\sqrt{2}}\left(|0\rangle -|1\rangle \right)\right\}$$Y-Basis:
$$\left\{|+i\rangle :=\frac{1}{\sqrt{2}}\left(|0\rangle +i|1\rangle \right),|-i\rangle :=\frac{1}{\sqrt{2}}\left(|0\rangle -i|1\rangle \right)\right\}$$

The probability that a state |ψ〉 collapses during a measurement in the basis $\left\{|b\rangle ,|b^{*}\rangle \right\}$, for the state |b〉 is provided by $P\left(b\right)={\left|\langle b|\psi \rangle \right|}^{2};\;\underset{i}{\Sigma }P\left(b\right)=1$, as claimed by the Born rule. For example, we measure the state $|\psi \rangle =\frac{1}{\sqrt{3}}\left(|0\rangle +\sqrt{2}|1\rangle \right)$ in the $\left\{|0\rangle ,|1\rangle \right\}$ basis,

$\;\;\;P\left(0\right)={\left|\langle 0|\frac{1}{\sqrt{3}}\left(|0\rangle +\sqrt{2}|1\rangle \right)\rangle \right|}^{2}={\left|\frac{1}{\sqrt{3}}\langle 0|0\rangle +\frac{\sqrt{2}}{\sqrt{3}}\langle 0|1\rangle \right|}^{2}=\frac{1}{3}$ and $P\left(1\right)=1-\frac{1}{3}=\frac{2}{3},$ where 〈0|0〉=1 (normalized) and 〈0|1〉=0 (orthogonal); thus, the qubit is in the state |1〉.

### 2.3. Quantum Circuit

Quantum circuits are represented using circuit diagrams. These diagrams are built and read from left to right. Barenco et al. [34] defined several of the basic operators that we now use in quantum circuits today. Two binary operators were added to this set by Toffoli and Fredkin [35,36]. We start building a quantum circuit with a line representing a circuit wire. As seen in Figure 1a, on the left side of the wire, a ket indicates its original condition of state preparation.

The qubit stays in the state in which it was originally created if there is no operator on the line. This indicates that the qubit state is maintained by the quantum computer.

The number of qubits produced in that condition is denoted by a slash *n* sign all over the wire as shown in Figure 1b.

A quantum circuit is a computer process that combines traditional real-time computation with coherent quantumgates on quantum data such as qubits.

### 2.4. Quantum Gates

Quantum gates or operators fundamentally involve the modification of one or more qubits. Single-qubit gates are represented as a box with the letter of the operator straddling the line. An operator box connecting two quantum wires is the basis of a binary gate.

#### 2.4.1. Single Qubit Gates

Pauli matrices are the first of three operators we examine, which are used to represent certain typical quantum gates.

Pauli-X Gate: Is the quantum equivalent of the NOT gate used in traditional computers, generally termed as the bit flip operator or x $\sigma_{x}$:
$$X:=\left(\begin{array}{cc} 0 & 1\\ 1 & 0 \end{array}\right).$$When we apply *X* to |0〉, we obtain
$$\left(\begin{array}{cc} 0 & 1\\ 1 & 0 \end{array}\right)\left(\begin{array}{c} 1\\ 0 \end{array}\right)=\left(\begin{array}{c} 0\\ 1 \end{array}\right)=|1\rangle .$$As we can see, this gate flips the amplitudes of the |0〉 and |1〉 states. In a quantum circuit, the symbol in Figure 2a represents the Pauli-X gate.Pauli-Y Gate: Is commonly abbreviated as $\sigma_{y}$, which transforms the state vector throughout the whole *y*-axis:
$$Y:=\left(\begin{array}{cc} 0 & -i\\ i & 0 \end{array}\right).$$Consequently, when it is applied to the |1〉 state, we obtain
$$\left(\begin{array}{cc} 0 & -i\\ i & 0 \end{array}\right)\left(\begin{array}{c} 0\\ 1 \end{array}\right)=\left(\begin{array}{c} -i\\ 0 \end{array}\right)=-i|0\rangle .$$In Figure 2b, the circuit design for the Y operator is displayed.Pauli-Z Gate: The Z operator, also known as the phase flip operator, used to perform a 180${}^{\circ }$ rotation of the state vector around the *z*-axis.
$$Z:=\left(\begin{array}{cc} 1 & 0\\ 0 & -1 \end{array}\right).$$By applying the Pauli-Z gate to the computational basis state, we obtain the result shown below:
$$Z|u\rangle ={\left(-1\right)}^{u}|u\rangle .$$For the particular case u=0, we present it in matrix form:
$$\left(\begin{array}{cc} 1 & 0\\ 0 & -1 \end{array}\right)\left(\begin{array}{c} 1\\ 0 \end{array}\right)=\left(\begin{array}{c} 1\\ 0 \end{array}\right)={\left(-1\right)}^{0}|0\rangle =|0\rangle .$$We have in the case where u=1,
$$\left(\begin{array}{cc} 1 & 0\\ 0 & -1 \end{array}\right)\left(\begin{array}{c} 0\\ 1 \end{array}\right)=\left(\begin{array}{c} 0\\ -1 \end{array}\right)={\left(-1\right)}^{1}|1\rangle =-|1\rangle .$$The Pauli-Z gate’s circuit diagram is shown in Figure 2c.Phase shift gates: Consists of a set of single-qubit gates which convert the basis states |0〉↦|0〉 and $|1\rangle \mapsto e^{i\varphi }|1\rangle$.The matrix below represents the phase shift gate:
$$P\left(\varphi \right)=\left(\begin{array}{cc} 0 & 1\\ 1 & e^{i\varphi } \end{array}\right),$$ where φ denotes the phase shift over a period of 2π. Typical instances include the
Pauli-Z gate where φ=π, *T* gate where $\varphi =\frac{\pi }{4}$, and *S* gate where $\varphi =\frac{\pi }{2}$.Hadamard Gate: This quantum operator is essential for quantum computing because it allows the qubit to transform from one computational basis state to a superposition.
$$H:=\frac{1}{\sqrt{2}}\left(\begin{array}{cc} 1 & 1\\ 1 & -1 \end{array}\right).$$When Hadamard gate is applied to the state |0〉, we obtain
$$\frac{1}{\sqrt{2}}\left(\begin{array}{cc} 1 & 1\\ 1 & -1 \end{array}\right)\left(\begin{array}{c} 1\\ 0 \end{array}\right)=\frac{1}{\sqrt{2}}\left(\begin{array}{c} 1\\ 1 \end{array}\right)=\frac{|0\rangle +|1\rangle }{\sqrt{2}},$$
and to state |1〉, we obtain
$$\frac{1}{\sqrt{2}}\left(\begin{array}{cc} 1 & 1\\ 1 & -1 \end{array}\right)\left(\begin{array}{c} 0\\ 1 \end{array}\right)=\frac{1}{\sqrt{2}}\left(\begin{array}{c} 1\\ -1 \end{array}\right)=\frac{|0\rangle -|1\rangle }{\sqrt{2}}.$$As can be seen, the Hadamard gate projects a computational basis state into the superposition of two states.

#### 2.4.2. Multi Qubit States and Gates

A single bit offers two possible states; in addition, a qubit state contains two complex amplitudes, as we saw before. Similarly, two bits might be in one of four states: 00, 10, 01, or 11. Four complex amplitudes are needed to present the state of two qubits. These amplitudes are stored in a 4D-vector:$$|u\rangle =u_{00}|00\rangle +u_{01}|01\rangle +u_{10}|10\rangle +u_{11}|11\rangle =\left[\begin{array}{c} u_{00}\\ u_{01}\\ u_{10}\\ u_{110} \end{array}\right].$$

The tensor product ⊗ can be used to characterize the combined state of two separated qubits:$$|u\rangle =\left[\begin{array}{c} u_{0}\\ u_{1} \end{array}\right];|v\rangle =\left[\begin{array}{c} v_{0}\\ v_{1} \end{array}\right],$$
$$|vu\rangle =|v\rangle ⊗|u\rangle =\left[\begin{array}{c} v_{0}\\ v_{1} \end{array}\right]\left[\begin{array}{c} u_{0}\\ u_{1} \end{array}\right]=\left[\begin{array}{c} v_{0}u_{0}\\ v_{0}u_{1}\\ v_{1}u_{0}\\ v_{1}u_{1} \end{array}\right].$$

States of this form (|v〉⊗|u〉) called uncorrelated, but there are also bipartite states which cannot be expressed as |v〉⊗|u〉; these states are correlated and sometimes entangled.

The CNOT or CX gate is one of several quantum gates which perform on multi qubits, if the first qubit is in state |1〉, the second qubit will execute the NOT operation, otherwise leaving it unaltered. Using the CNOT gate, we entangle two qubits in a quantum circuit. This gate is described by the following matrix:$$CNOT=\left(\begin{array}{cccc} 1 & 0 & 0 & 0\\ 0 & 1 & 0 & 0\\ 0 & 0 & 0 & 1\\ 0 & 0 & 1 & 0 \end{array}\right).$$

As an instance, the CNOT gate may be applied to the state |10〉 as follows:$$\left(\begin{array}{cccc} 1 & 0 & 0 & 0\\ 0 & 1 & 0 & 0\\ 0 & 0 & 0 & 1\\ 0 & 0 & 1 & 0 \end{array}\right)\left(\begin{array}{c} 0\\ 0\\ 1\\ 0 \end{array}\right)=\left(\begin{array}{c} 0\\ 0\\ 0\\ 1 \end{array}\right)=|11\rangle .$$

The Table 1 reveals that the output state of the targeted qubit matches that of a conventional XOR gate. The targeted qubit is |0〉 when both inputs are the same, but |1〉 when the inputs are different. We represent the CNOT gate with the circuit’s diagram in Figure 2d.

### 2.5. Representation of Qubit States

The state of a qubit may be represented with different ways. Dirac notation allows us to express this state in a readable form. A qubit in state |0〉, for instance, will transfer to state |1〉 after the application of the X operator.
X|0〉→|1〉

In Figure 3, a state of a single qubit is represented by the Bloch sphere. Quantum states are represented by vectors that extend from the origin to a certain point on the surface of the Bloch sphere. The top and bottom antipodes of the sphere are |0〉 and |1〉, respectively.

On the Bloch sphere, we may write any purestate (i.e., a qubit state specified by a vector of norm 1. Mixedstate, in contrast, is a state that combines numerous pure quantum states or qubits) as seen below:(8)$$|\psi \rangle =\cos\left(\theta /2\right)|0\rangle \;+\;e^{i\phi }\sin\left(\theta /2\right)|1\rangle ,$$
with θ∈[0,π], determines the probability to measure the state |0〉 as $P\left(0\right)=\cos^{2}\left(\theta /2\right)$ and the state |1〉 as $P\left(1\right)=\sin^{2}\left(\theta /2\right)$, and ϕ∈[0,2π] describes the relative phase. On the surface of a Bloch sphere, all of these pure states may be represented with a radius of $|\vec{r}|=1$. The Bloch vector generates such a state’s coordinates:(9)$$\vec{r}=\left(\begin{array}{c} \sin(\theta )\cos(\phi )\\ \sin(\theta )\sin(\phi )\\ \cos(\theta ) \end{array}\right),$$
as result,

|0〉:θ=0,ϕ arbitrary $\to \vec{r}=\left(0,0,1\right)$;|1〉:θ=π,ϕ arbitrary $\to \vec{r}=\left(0,0,-1\right)$;$|+\rangle :\theta =\pi /2,\phi =0\to \vec{r}=\left(1,0,0\right)$;$|-\rangle :\theta =\pi /2,\phi =\pi \to \vec{r}=\left(-1,0,0\right)$;$|+i\rangle :\theta =\pi /2,\phi =\pi /2\to \vec{r}=\left(0,1,0\right)$;$|-i\rangle :\theta =\pi /2,\phi =3\pi /2\to \vec{r}=\left(0,-1,0\right)$.

Bloch sphere is only capable of representing the state of a particular qubit. Therefore, the Q−sphere is used for multi-qubits (and single qubits as well).

For a single qubit as shown in Figure 4a:-The south pole represents the state |1〉;-The north pole represents the state |0〉;-The size of the blobs is related to the likelihood that the relevant state will be measured;-The color indicates the relative phase compared to the state |0〉.For *n* qubits:In Figure 4b, we plot all states as equally distributed points on sphere with $0^{⊗n}$ on the north pole, $1^{⊗n}$ on the south pole, and all other state are aligned on parallels, such as the number of “1”s on each latitude is constant and increasing from north to south.

### 2.6. Entanglement

Quantum entanglement is a phenomena wherein quantum particles interact and are represented by reference to one another despite their great distance apart. At the point of measurement, if one of the entangled particles in a pair is determined to be in the ’down’ spin state, this state is instantly transmitted to the other particle, which now adopts the opposite spin, state of ’up’. Even if entangled particles were positioned on opposite corners of the universe, they would remain “connected”. This example is significant because it demonstrates how wave–particle duality qualities may be used to allow qubits to interact with quantum algorithms via interference in some cases. We consider, for example, the state $\frac{1}{\sqrt{2}}\left(|00\rangle +|11\rangle \right)$ (which is a Bellstate), and this state has a chance of 50 percent of being measured in state |00〉 and another half chance of being measured in the state |11〉. The measurement of one causes the superposition to collapse and seems to have an instantaneous impact on the other.

The Bell states, also called EPR pairs, are quantum states with two qubits that represent the simplest (and most extreme) forms of quantum entanglement; they are a form of normalized and entangled basis vectors. This normalization indicates that the particle’s overall probability of being in one of the states stated is 1: 〈ψ|ψ〉=1 [38].

Four fully entangled states known as Bell states exist, and they constitute an orthonormal basis (i.e, a basis in which all of the vectors have the same unit norm and are orthogonal to one another):1.$|\psi^{00}\rangle =\frac{1}{\sqrt{2}}\left(|00\rangle +|11\rangle \right)$.2.$|\psi^{01}\rangle =\frac{1}{\sqrt{2}}\left(|01\rangle +|10\rangle \right)$.3.$|\psi^{10}\rangle =\frac{1}{\sqrt{2}}\left(|00\rangle -|11\rangle \right)$.4.$|\psi^{11}\rangle =\frac{1}{\sqrt{2}}\left(|01\rangle -|10\rangle \right)$.

Although there are a variety of ways to create entangled Bell states using quantum circuits, the most basic uses a computational basis as input and includes a Hadamard gate and a CNOT gate (see Figure 5a).

The circuit calculation results are displayed in Table 2.

In the opposite sense: Bell’s measure is a standard measure with standard outputs *i* and *j* corresponds to a measure of the state $|\psi_{ij}\rangle$. Figure 5b represents the circuit of Bell measurement.

### 2.7. Quantum Computer

Quantum computers are processing machines that take advantage of quantum physics properties. This may be extremely beneficial for some tasks, where they can greatly outperform the most powerful supercomputers. A quantum computer can be considered as a co-processor of a traditional computer, as a GPU can be for video games or for training neural networks in deep learning. As shown in Figure 6, a traditional computer closely control computer operations by triggering at precise rates the qubit operations performed by the quantum gates. This trigger takes into consideration the execution time of the quantum gates as well as the known coherencetime of the qubits, i.e., the duration for which the qubits remain in a superposition state. In addition to its classical control computer, the quantum computer includes several components, which we analyze one by one, with the details below:Quantum registers are just a collections of qubits. In November 2022, the benchmarked record was 433 qubits, announced by IBM. Quantum registers store the information manipulated in the computer and exploit the principle of superposition allowing a large number of values to coexist in these registers and to operate on them simultaneously;Quantum gates are physical systems acting on the qubits of the quantum registers, to initialize them and to perform computational operations on them. These gates are applied in an iterative way according to the algorithms to be executed;At the conclusion of the sequential execution of quantum gates, the measurement interface permits the retrieval of the calculations’ results. Typically, this cycle of setup, computation, and measurement is repeated several times to assess the outcome. We then obtain an average value between 0 and 1 for each qubit of the quantum computer registers. The values received by the physical reading devices are therefore translated into digital values and sent to the traditional computer, which controls the whole system and permits the interpretation of the data. In common cases, such as at D-Wave or IBM, which are the giants of quantum computers building, the calculation is repeated at least 1024 times in the quantum computer;Quantum chipset includes quantum registers, quantum gates and measurement devices when it comes to superconducting qubits. Current chipsets are not very large. They are the size of a full-frame photo sensor or double size for the largest of them. The size of the latest powerful quantum chip called the 433-qubit Osprey, around the size of a quarter;Refrigerated enclosure generally holds the inside of the computer at temperatures near absolute zero. It contains part of the control electronics and the quantum chipset to avoid generating disturbances that prevent the qubits from working, especially at the level of their entanglement and coherence, and to reduce the noise of their operation;Electronic writing and reading in the refrigerated enclosure, control the physical devices needed to initialize, update, and read the state of qubits.

Quantum computers today are built based on atoms (e.g., cold atoms, trapped ions and nuclear magnetic resonance), electrons (e.g., superconducting, silicon and topological), or photon (e.g., linear optics). D-Wave, IBM and Google have all shown significant quantum computer development progress. The D-Wave model is built using the quantum adiabatic technique, which handles issues involving optimization or probabilistic sampling, whereas the IBM model is mostly dealing with non-adiabatic models. Most of those companies provide a quantum computer simulator that runs on a classical computer to test code before executing it on a quantum device. It is possible to use this simulator in either a local environment or the cloud. It cannot handle true quantum states because it is operating on a classical computer, but it is useful for testing code syntax and flow.

### 2.8. Quantum Algorithms

The creation of quantum algorithms necessitates a higher level of expertise than traditional algorithms and programs. Quantum computers will need the development of a new generation of mathematicians and developers capable of reasoning with the mathematical formalization of quantum programming. Furthermore, these algorithms must be more efficient than those designed for traditional computers or supercomputers. Given that quantum computing employs a distinct method of computation, it is only reasonable to ask what kinds of problems may now be solved in this new environment, even if they were not expected to be solved in a traditional computer. To do this, we must review the theory of complexity.

The complexity theory focuses on the scalability and computational cost of algorithms in general and for specific problems. Scalability refers to the time and/or space necessary to increase the volume or complexity of a computation’s goal. Using the Big-O notation, an algorithm that is $O(n^{3})$ is considered to be “harder” than one that is $O(n^{2})$, since the former will often affect more operations than the latter, regardless of the time at which these operations are executed. A specific complexity class is composed of problems that share similar characteristics of difficulty. The most significant categories of complexity are discussed below:Polynomial time (P): Are issues solvable in a polynomial amount of time. In other terms, a traditional computer is capable of resolving the issue in a reasonable time;Non-deterministic Polynomial time (NP): Is a collection of decision issues that a nondeterministic Turing machine might solve in polynomial time. P is an NP subset;NP-Complete: X is considered to be NP-Complete only if the requirements here are met: (i) X is in NP, and (ii) in polynomial time, all NP problems are reducible to X. We assert that X is NP-hard if only (ii) is true and not necessarily (i);Polynomial Space (PSPACE): This category is concerned with memory resources instead of time. PSPACE is a category of decision issues that can be solved by an algorithm whose total space utilization is always polynomially restricted by the instance size;Bounded-error Probabilistic Polynomial time (BPP): Is a collection of decision issues which may be handled in polynomial time using a probabilistic Turing computer with such a maximum error probability equal to 1/3;Bounded-error Quantum Polynomial time (BQP): A decision issue is BQP, if it has a polynomial time solution and has a high accuracy probability. BQP is the basic complexity category of problems which quantum computers may effectively solve. It corresponds to the classical BPP class on the quantum level;Exact Quantum Polynomial time (EQP or QP): Is a type of decision issue that a quantum computer can handle in polynomial time with probability 1. This is the quantum counterpart of the P complexity class.

The potential uses of quantum computing vary from decrypting cryptography systems to the creation of novel medications. These applications use quantum algorithms, which are programs that exploit quantum gates on a quantum computer in order to obtain a speedup or even other advantages over traditional algorithms. An important consideration in the creation of quantum algorithms is to ensure that they are more efficient than their counterparts optimized for traditional computers. Theories exist to verify this by evaluating the exponential, polynomial, logarithmic, or linear scaling of the computation time as a measure of the task’s complexity, or a combination of all four. Currently, there are four primary classes of quantum algorithms: searchalgorithms based on those of Deutsch Jozsa [39] and Grover [1]; algorithmsbasedonQuantumFourierTransforms (QFT) such as Shor’s algorithm [2], which is used to factor integers; quantumannealingalgorithms, algorithms that search for an equilibrium point of a complex system as in neural network training, optimal path search in networks or process optimization; and quantumsimulationalgorithms, which are used to simulate the interactions between atoms in various molecular structures. Quantum chemistry, which simulates the impact of chemical stimulation on a huge amount of atomic particles, is one particularly exciting topic in quantum simulation.

One of the first quantum algorithms invented is that of David Deutsch, called Deutsch Jozsa algorithm [39], co-invented with Richard Jozsa. This algorithm allows for identifying the function of a “black box” called an “oracle” (a black box that is frequently used in quantum algorithms to estimate functions using qubits), which we know in advance that it will return for all its inputs, either always the same value, 0 or 1, or the values 0 and 1 in equal parts. The algorithm thus makes it possible to know if the function has a balanced output. It is implemented on a set of qubits n, all of the input qubits are set to zero, except one which is initialized to 1, then they are each put in superposition between 0 and 1 using Hadamard gates. The qubits thus have simultaneously all the possible values with $2^{n+1}$ combinations of values. It is easy to understand why this quantum algorithm is much more efficient than its conventional counterpart. In conventional computation, more than half of the possible input values would have to be sequentially scanned, whereas, in the quantum version, they are all scanned at the same time. This is an example of a strong algorithm that has no known practical use. Moreover, there are classical probabilistic algorithms algorithms that erase a good part of the quantum power gain of the Deutsch–Jozsa algorithm.

Grover’s algorithm is the other popular algorithm [1], created in 1996. It allows for a fast quantum search in a database. A bit like the Deutsch–Jorza algorithm, it scans a list of elements to find those that verify a specific condition. It also uses the superposition of qubit states to speed up the processing compared to a traditional sequential search. The improvement in performance is significant compared to an unsorted database. Grover’s algorithm also uses an “oracle” or “black box” function that will indicate whether a set of input qubits verifies a search condition or not, such as verifying that a given phone number has been found in a list of phone numbers. In such a case, the function compares the phone number searched and the one submitted to it, to answer one if they are identical and zero otherwise. The black box is a quantum box and will evaluate this function for 2N registers of qubits at the same time. It will therefore output a one once and zeros elsewhere. The computing time is determined by the square root of the list size, while a classical approach has a computation time proportional to the size of the list. Therefore, going from a time *N* to $\sqrt{N}$ is an interesting gain. This algorithm can then be used to be integrated in other algorithms such as those which allow for discovering the optimal path in a graph or the minimal or maximal number of a series of *N* numbers.

The quantum Fourier transform (QFT) was invented by Don Coppersmith in 1994 [40]. It is used in various other algorithms and in particular in Shor’s algorithm, which is a factoring algorithm for integers. The QFT could compute the prime factorization of an N-bit integer in $\log^{2}\left(N\right)$ time, whereas the most well-known basic Fourier transform requires N×log(N) time. IBM performed one of the initial implementations of Shor’s technique in 2001 using an experimental quantum computer with seven qubits to factor the integer 15. In 2012, Nanyang et al. [41] claimed factorization of 143 using an NMR adiabatic quantum computer at ambient temperature (300 K). Using a D-Wave 2X quantum computer, the 18-bit number 200 099 successfully factored in April 2016 utilizing quantum annealing [42]. Late in 2019, an industry collaboration determined that 1,099,551,473,989 equals 1,048,589 times 1,048,601 [43]. Note that Shor’s algorithm also allows for breaking cryptography using elliptic curves [44], which compete with RSA cryptography. Incidentally, a part of the cryptography used in the Bitcoin protocol would also pass through the Shor algorithm [45]. In addition to the purely quantum algorithms (based only on quantum gates), we must indeed add: hybrid algorithms, combining traditional algorithms and quantum algorithms or algorithms based on quantum-classical gates. This is notably the situation involving the Variational Quantum Eigensolver (VQE) [46], which allows the solution of chemical simulation problems as well as neural network training; and quantum inspired algorithms, which are algorithms for classical computers inspired by quantum algorithms for solving complex problems.

## 3. Quantum Machine Learning

There are four strategies of how to combine machine learning and quantum computing, based on if one considers the data was created by a classical (C) or quantum (Q) system, and whether the computer that processes data are classical (C) or quantum (Q),(as shown in Figure 7).

The scenario CC refers to data that is treated traditionally. This is the traditional technique of machine learning; however, in this case, it refers to machine learning using quantum information research-derived approaches. Tensor networks that are designed addressing to quantum many-body processes, have been employed in training neural networks [48]. In addition, there are various ’quantum-inspired’ machine learning algorithms with varied degrees of quantum theoretical background.

The scenario QC examines how machine learning might aid quantum computing. For instance, if we wish to obtain a complete explanation of the state of a computing device by taking a few measurements, we may analyze the measurement data using machine learning [49]. There are several applications for machine learning to distinguish either quantum states generated by a quantum source or manipulations carried out by a quantum experiment [50,51].

The scenario CQ uses quantum computing to examine conventional datasets. Observations from classical computers, which include images and text, are supplied to a quantum device for evaluation. The primary objective of the CQ methodology is the development of quantum algorithms to be used in data mining, for which the community has offered a variety of solutions. They might be adaptations of traditional machine learning models to function with quantum algorithms, or they can be entirely novel creations inspired by the characteristics of quantum computing.

The last scenario QQ examines the processing of “quantum data” by a quantum device. There are two possible interpretations of this. The information may be obtained through a quantum experiment involving a quantum system and the subsequent input of those experimental measurements into a different quantum computer. In a second interpretation, a quantum computer simulates the behavior of the quantum system and then uses the state of this system as such an input of a QML algorithm performed on the same computer.

Nowadays, Quantum Machine Learning (QML) algorithms are separated into three distinct strategies. Quantum machine learning algorithms, which are quantum versions of classical machine learning, in addition to algorithms that are implemented on a real quantum computer, including QSVM [4], Quantum Neural Network [52] and Quantum Linear Regression [8]. The second strategy is quantum-inspired machine learning, which leverages the concepts of quantum computing to enhance traditional machine learning algorithms. A few examples of such algorithms include quantum-inspired binary classifier [53], Helstrom Quantum Centroid which is a quantum-inspired binary supervised learning classification [54], Quantum-inspired Support Vector Machines [55], Quantum Nearest Mean Classifier [56], inspired Quantum K Nearest-Neighbor [57], quantum algorithms for ridge regression [58] and Quantum inspired Neural network [59]. The third strategy, hybrid classical-quantum machine learning algorithms, merges classical and quantum algorithms to improve performance and reduce the cost of learning—for example, using quantum circuit to propose a novel variational quantum classifier [60], variational quantum SVM and SVM quantum kernel-based algorithm [61], quantum KNN algorithm [62], and a hybrid quantum computer-based quantum version of nonlinear regression [63].

In our study, we concentrate on the CQ scenario. There are two distinct approaches to the development of quantum machine learning models. The first approach involves running traditional machine learning algorithms on quantum computers or simulators in an effort to achieve algorithmic speedups. This approach needs to translate conventional data into quantum data, a process known as quantumencoding. Another approach is to create QML algorithms based on quantum subroutines, including Grover’s algorithm, HHL algorithm, quantum phase estimation, and Variational Quantum Circuit (VQC), which we cover later in detail.

Figure 8 illustrates the processing strategies used in conventional machine learning and QML. In traditional machine learning, data are a direct input to the algorithm, which then analyses the data and generates an output. QML, on the other hand, demands the initial encoding of the data into quantum data. QML receives quantum data as input, processes it, and generates quantum data as output. The quantum data are then converted to conventional data.

Quantum Machine Learning requires a more intricate encoding of classical to quantum data than classical machine learning.

### 3.1. Quantum Encoding

The majority of machine learning models depends on sets of data with ${\left\{x_{i}\right\}}_{i}\in \left[N\right]$ samples, where $x_{i}\in R^{d}$ for every *i*. Additionally, one requires classical outcomes including labels or classes ${\left\{y_{i}\right\}}_{i}\in \left[N\right]$, where $y_{i}\in {R^{d}}^{{}^{\prime }}$ for every *i*. It appears that the ability to deal with conventional data will be required for the vast majority of practical applications of quantum machine learning [64].

In a Hilbert space, there are several strategies for encoding data from conventional data to quantum data. In other words, data encoding requires putting conventional data into a quantum computer as quantum states. We describe three methods. We recommend [65] for information on different strategies.

#### 3.1.1. Basis Encoding

Basis encoding links a computational basis state for an n-qubit system. Therefore, traditional data should be the binary strings, and encoded quantum state is the bit-wise conversion of a binary string towards the equivalent states of the quantum subsystems. In example, x=1001 is encoded by the 4-qubit quantum state |1001〉. Furthermore, one bit of classical data are encoded by one quantum subsystem. In some ways, it is the most basic type of computing since every bit is effectively replaced with a qubit, and computations are carried out simultaneously on all bits in a superposition. Consequently, the bit encoding for a vector is the following: m+[log(d)] qubits could a vector $x=\left(x_{1},\ldots ,x_{d}\right)\in R^{d}$ as a quantum superposition of bit strings, where each instance must be a binary string with *N* bits for the basis encoding. $x_{i}=\left(b_{1},\ldots ,b_{j},\ldots ,b_{N}\right)$ for j=1,…,N with $b_{j}\in \left\{0,1\right\}$:(10)$$|x\rangle =\frac{1}{\sqrt{d}}\sum_{i=1}^{d}|x_{i}\rangle ,$$
where *m* denotes the amount of qubits used for the precision, while *d* represents the number of samples. To encode a vector x=(2,3) in basis encoding. First, we must transform it to a binary format, with 2 bits, $x_{1}=10$, $x_{2}=11$. The corresponding basis encoding makes use of two qubits to represent the data as below:$$|x\rangle =\frac{1}{\sqrt{2}}|10\rangle +\frac{1}{\sqrt{2}}|11\rangle .$$

#### 3.1.2. Amplitude Encoding

To employ qubits to represent classical vectors or matrices, the most efficient encoding is by far the amplitude encoding approach. It is the theoretical link between quantum computing and linear algebra that uses quantum features to the maximum. The amplitudes of an n-qubit quantum state $|\psi_{x}\rangle$ as $|\psi_{x}\rangle =\sum_{i=1}^{d}x_{i}|i\rangle ,$ with $d=2^{n}$ represent a normalized classical d-dimensional set of data *x*, where $x_{i}$ is the *i*-th item, and |i〉 is the *i*-th computational base state. However, in this case, $x_{i}$ might be any numeric value, such as a floating or integer point. For instance, suppose we want to use amplitude encoding to encode the four-dimensional array x=(1.0,3.0,−2.5,0.0). The first step is to normalize this array, which is achieved by setting $x_{norm}=\frac{1}{\sqrt{16,25}}\left(1.0,3.0,-2.5,0.0\right)$. Two qubits are used to encode $x_{norm}$ in the corresponding amplitude encoding as follows:$$|\psi_{x_{norm}}\rangle =\frac{1}{\sqrt{16,25}}\left[|00\rangle +3|01\rangle -2.5|10\rangle \right].$$

This result state will at the same moment encode the matrix below:$$X=\frac{1}{\sqrt{16,25}}\left(\begin{array}{cc} 1.0 & 3.0\\ -2.5 & 0.0 \end{array}\right),$$
where the first qubit represents the index for such a row, while the second qubit represents the index for such a column.

#### 3.1.3. Qsample Encoding

Qsample encoding is a hybrid scenario of amplitude and basis encoding, and it combines a real amplitude vector with conventional binary probability distributions. Suppose an amplitude vector $v={\left(v_{1},\ldots ,v_{2^{n}}\right)}^{T}$ that describes an n-qubit quantum state. Sampling a series of bits of size *n* with a set of discrete probabilities $p_{1}={\left|v_{1}\right|}^{2}$,…, $p_{2^{n}}={\left|v_{2^{n}}\right|}^{2}$ is comparable to measuring the *n* qubits. This means that the amplitude vector may be used to characterize the distribution of an a classical discrete random variable [66]. Taking into account a discrete classical probability distribution over binary representation $p_{1},\ldots ,p_{2^{n}}$, the quantum state is:(11)$$|\psi \rangle =\sum_{i=1}^{2^{n}}\sqrt{p_{i}}|i\rangle .$$

### 3.2. Essential Quantum Routines for QML

In this subsection, we discuss a few of the most basic quantum algorithms which aid in the development of quantum machine learning algorithms.

#### 3.2.1. HHL Algorithm

Using a quantum implementation, the Harrow–Hassidim–Lloyd algorithm [5] consists of solving a linear equation set. Solving a system of linear equations is comparable to the matrix inversion issue. The objective of the matrix inversion problem, with a matrix *M* and a vector *v*, is to determine the vector *x*:(12)Mx=v.

Numerous algorithms for machine learning determine their parameter θ through solving the Mθ=v matrix inversion issue, which makes this algorithm essential for quantum machine learning. In general, the matrix *M* represents the input characteristics of the training data points expressed in the matrix *X*. The data point matrix *X* and the targeted vector *Y* are both used to create the training data point vector *v*. In linear regression, in which the predicted output is $y=\theta^{T}x$, determining θ is as simple as solving the matrix inversion issue presented below:(13)$$\left(X^{T}X\right)\theta =X^{T}Y,$$
thus, $M=X^{T}X$ and $v=X^{T}Y$ in linear regression.

For the HHL algorithm, we must identify one or more operators capable of transforming the state |v〉 into the solution vector θ. Clearly, $M=X^{T}X$ would have to be included into one of the operators. If *M* is unitary, we cannot use it as the quantum operator. However, if *M* is Hermitian, we might assign it to a Hamiltonian denoted by *H* of a quantum system. As a reminder, a linear operator or matrix *H* is Hermitian only if its complex conjugate transpose $H^{†}$ is identical to *H*. Although if *M* is not Hermitian, a Hermitian operator $\tilde{M}$ may be defined as shown below:(14)$$\tilde{M}=\left[\begin{array}{cc} 0 & M^{†}\\ M & 0 \end{array}\right].$$

While $\tilde{M}$ is now Hermitian, it has the following eigenvalue decomposition:(15)$$\tilde{M}=\sum_{i}\lambda_{i}|u_{i}\rangle \langle u_{i}|,$$
in which the eigenvectors $|u_{i}\rangle$ provide an orthonormal basis. The vector |v〉 may be expressed as follows in the orthonormal basis $|u_{i}\rangle$:(16)$$|v\rangle =\sum_{i}\beta_{i}|u_{i}\rangle .$$

The following is the answer to the inverse problem:(17)$$|x\rangle ={\tilde{M}}^{-1}|v\rangle .$$

Despite the fact that $\tilde{M}$ is a Hermitian matrix of spectral decomposition $\tilde{M}=\sum_{i}\lambda_{i}|u_{i}\rangle \langle u_{i}|$, its inverse is given as follows:(18)$${\tilde{M}}^{-1}=\sum_{i}\frac{1}{\lambda_{i}}|u_{i}\rangle \langle u_{i}|.$$

The answer |x〉 is obtained by putting the value of ${\tilde{M}}^{-1}$ of Equation (Equation 18) as well as |v〉 of Equation (Equation 16) into Equation (Equation 17), which can be seen below:(19)$$|x\rangle =\sum_{i}\frac{1}{\lambda_{i}}|u_{i}\rangle \langle u_{i}|\sum_{j}\beta_{j}|u_{j}\rangle =\sum_{j}\frac{\beta_{j}}{\lambda_{j}}|u_{j}\rangle .$$

Equation (Equation 19) demonstrates that, if we could move from eigenstates $|u_{i}\rangle$ to $\frac{1}{\lambda_{i}}|u_{i}\rangle$, then would indeed be nearer to the answer. Quantumphaseestimation (which we discuss later) is one approach for achieving this objective, which used the unitary operator $U=e^{-i\tilde{M}t}$ upon that state |v〉 represented as the superposition state of such basis states $|u_{i}\rangle$, as this will transform the eigenstates $|u_{i}\rangle$ into $\lambda_{i}|u_{i}\rangle$. Then, the eigenvalues may be inverted from $\lambda_{i}|u_{i}\rangle$ to $\frac{1}{\lambda_{i}}|u_{i}\rangle$ by a controlled rotation. Let us remember that the state |v〉 must have a unit norm before a quantum phase estimate could be performed on it. In compared to its classical version, the algorithm asserts exponential gains in speedup for certain issue formulations. The result of the procedure requires $O({\left(logN\right)}^{2})$ quantum states, but O(NlogN) conventional computing processes.

#### 3.2.2. Grover’s Algorithm

The speed at which database components may be accessed is one of the possible benefits of quantum computing above traditional computing. Such an algorithm is Grover’s algorithm, which may yield a quadratic speedup while searching a database. The algorithm of Grover [1] employs the amplitude amplification approach [67] that is used in database search as well as in several other applications.

Assuming we have such a database containing $N=2^{n}$ entries, and we wish to find the one indexed by *w*, which we refer to as the winner. The computational basis states |x〉 that correspond to *n* input quantum bits may be used to define the *N* items. The oracle operates on every one of the computational basis states $|x\rangle \in {\left\{0,1\right\}}^{n}$ with the output f(x)=1, while for the other elements, it returns 0. In the paradigm of quantum computing, the oracle $U_{w}$ may be seen as the unitary operator which further operates upon that computational basis state |x〉, seen below:(20)$$U_{w}|x\rangle ={\left(-1\right)}^{f(x)}|x\rangle .$$

The Oracle has the following impact on the winning element indicated element |w〉 specified by the computational basis state:(21)$$U_{w}|w\rangle ={\left(-1\right)}^{f(w)}|w\rangle =-1^{1}|w\rangle =-|w\rangle .$$

Now that we are already familiar with the oracle, let us examine the stages of Grover’s method. Let |w〉 represent the superposition of every state:(22)$$|s\rangle =\frac{1}{\sqrt{N}}\sum_{x=0}^{N-1}|x\rangle ,$$
and the operator,
(23)$$U_{s}=2|s\rangle \langle s|-I,$$
is called the Grover diffusion operator. We can consider the winner |w〉 and an extra state $|s^{\prime }\rangle$ which is in the span of |w〉 and |s〉, which is orthogonal to |w〉, and which is obtained from |s〉 by eliminating |w〉 and rescaling.

1.Let us just put the system in the state |s〉,
(24)$$|s\rangle =\frac{1}{\sqrt{N}}\sum_{x=0}^{N-1}|x\rangle .$$This superposition |s〉, which is easily produced from $|s\rangle =H^{⊗n}{|0\rangle }^{n}$, is the beginning for the amplitude amplification technique, as shown in Figure 9.The left chart corresponds to the two-dimensional plane spanned by orthogonal vectors |w〉 and $|s^{\prime }\rangle$ which allows for describing the beginning state as $|s\rangle =\sin\theta |w\rangle +\cos\theta |s^{\prime }\rangle$, where $\theta =\arcsin\langle s|w\rangle =\arcsin\frac{1}{\sqrt{N}}$. The right picture is a bar chart of the amplitudes of the state |s〉.2.Execute r(N) times the following “Grover iteration”:(a)Apply the $U_{w}$ operator to |s〉.Geometrically, this relates to a reflection of the state |s〉 about $|s^{\prime }\rangle$. This transformation indicates that the amplitude in front of the |w〉 state turns negative, which in turn implies that the average amplitude (shown by a dashed line in Figure 10) has been reduced.(b)We now implement the operator $U_{s}$ to the state |s〉.This transformation completes the transformation by matching the state to $U_{s}U_{w}|s\rangle$, which relates a rotation around an angle θ as shown in Figure 11.The state will rotate by r×θ after *r* implementation of step 2, where $r=\frac{\pi }{4}\sqrt{2^{n}}\equiv O\left(\sqrt{N}\right)$ [38].3.The final measurement will give the state |w〉 with probability $P\left(\omega \right)\geq 1-\sin^{2}\left(\frac{\theta }{2}\right)=1-\frac{1}{2^{n}}$.

#### 3.2.3. Quantum Phase Estimation

Quantum phase estimation is a quantum algorithm that employs the Quantum Fourier Transform (QFT) to convert information encoded within the phase φ with an amplitude $\alpha =\left|\alpha \right|e^{i\varphi }$ of a state. The QFT accelerates exponentially the process of translating a quantum state encoded vector into Fourier space. It is often utilized in QML algorithms to retrieve the information contained in the eigenvalues of operators which includes details of data points. Phase estimation consists essentially to identify the eigenvalues of such a matrix *U*, which is represented as the operator *U* in the quantum circuit *U*, hence this operator must be unitary. We designate their eigenvectors by $|u_{j}\rangle$ as well as eigenvalues $e^{i\theta j}$, thus $U|u_{j}\rangle =e^{i\theta j}|u_{j}\rangle$. Assuming as inputs an eigenvector and an additional register $|u_{j}\rangle |0\rangle$, the method should return $|u_{j}\rangle |\theta_{j}\rangle$. Because the eigenvectors form a basis, it is possible to express every state |ψ〉 as $|\psi \rangle =\sum_{j\in [n]}\alpha_{j}|u_{j}\rangle$. Phase estimation is therefore particularly convincing due to its potential for use in superposition. The circuit diagram is shown below. Given an eigenvector as well as an extra register $|u_{j}\rangle |0\rangle$ as input, the algorithm will return $|u_{j}\rangle |\theta_{j}\rangle$. Since eigenvectors constitute a basis, each state |ψ〉 could be represented as $|\psi \rangle =\sum_{j\in [n]}\alpha_{j}|u_{j}\rangle$. Thus, the most intriguing aspect of phase estimation is its application in superposition. The circuit diagram of the quantum phase estimation is shown in Figure 12.

One has the option of measuring the final output to obtain a classical description of the eigenvalues or retaining it as a quantum state for future computing. Recall that this circuit requires the sequential controlled application of the unitary *U*. For such *U* circuits, this can be an expensive operation.

#### 3.2.4. Variational Quantum Circuit

Variational quantum circuits (VQC) [68] is the method that has attracted the most interest from researchers. It was suggested using the variational quantum eigensolver (VQE) method for quantum chemistry [46] and the quantum approximation optimization algorithm (QAOA) [31] for optimization based on classical machine learning. The latter used for a lot of related applications in machine learning [27,69,70]. Variational quantum circuit have universal qualities [71,72] and have already shown positive outcomes in real-world tests [73], but their natures are vastly different. They generally based on the approach shown in Figure 13. The ansatz is a tiny circuit comprised of numerous gates having adjustable characteristics, including the angle of a gate that controls rotation. The resultant quantum state is then measured, which should provide the correct answers for the intended problem (classification, regression). Initial findings are poor since settings are almost random. The name of this measure referred to as the Loss or the Objective Function. In the end, an optimization is performed on a traditional computer in order to provide a hopefully improved set of parameters for the experiment to test. In addition, this process is repeated until the circuit produces acceptable results.

When performing a VQC, it is necessary to make an estimate of the gradients of such a cost function relative to every parameter. In conventional neural networks, this is typically achieved through backpropagation across analytic procedures. While using VQC, processes become excessively complicated, and we are unable to reach intermediate quantum states unless we first measure them. In the present day, the progressed approach is referred to as the parameter shift rule [73], and needs to apply the circuit twice for each parameter and measuring its outcome twice. In contrast, traditional deep learning just requires a single forward and backward pass through the network to collect all of the thousand of gradients. The parameter shift rule could be parallelizable over several simulators or quantum devices. However, this might be impractical for a high number of parameters.

Table 3 presents a comprehensive overview of the current QML algorithms and indicates the quantum routines they are based on or have used.

### 3.3. QML Algorithms

#### 3.3.1. Quantum Support Vector Machines

Support Vector Machines, generally known as SVMs, are a type of supervised algorithm for machine learning that may be used to handle problems involving linear discrimination. The approach involves establishing a hyperplane that differentiates between two different classes of feature vectors. This hyperplane serves as a decision threshold for the further categorization of data and is based on the concept of finding the hyperplane. The SVM is described as aiming to maximize the distance between the hyperplane and the support vectors of the data points that are located nearest to it. Depending on the kernel employed by the SVM method, the objective function is sometimes convex or not. Non-convex functions are often closer to the local maximum; hence, the conventional SVM compromises optimization efficiency and accuracy rate. QSVM uses Grover’s algorithm as a quantum subroutine for reduction. This makes it possible for non-convex cost functions to converge to their global optimal value [79]. We describe the Quantum SVM algorithm as the following. For clarity, consider linear SVMs with hyperplanes given below:(25)$$\theta^{T}x-c=\pm 1.$$

Recall that *c* is a constant value while *x* and θ are vectors. If θ is constrained to a minimum, then the margin may be relied upon to accurately categorize the data. All of that is characterized by the following optimization issue:(26)$$\underset{\theta ,c}{min}\frac{1}{2}{\left||\theta |\right|}^{2},$$
subject to one constraint:(27)$$y^{i}\left(\theta^{T}x^{\left(i\right)}-c\right)\geq 1.$$

In the training data i={1,…,M} and $y^{\left(i\right)}=\left\{-1,1\right\}$, using Lagrange multipliers $\alpha^{\left(i\right)}$, the constraint may be included into an objective function, as a consequence, and the problem might be described as follows:(28)$$\underset{\theta ,c}{min}\underset{\alpha^{\left(i\right)}\geq 0}{max}\left(\frac{1}{2}{\left||\theta |\right|}^{2}-\sum_{i=1}^{M}\left[\alpha^{\left(i\right)}\left(\theta^{T}x^{\left(i\right)}-c\right)-1\right]\right).$$

It is important to note that the only non-zero value of $\alpha^{\left(i\right)}$ is equivalent to the sum of the support vectors $x^{\left(i\right)}$. The derivatives listed below are all set to zero order to maximize the objective function F with regard to $\alpha^{\left(i\right)}$:(29)$$\frac{\partial F}{\partial \theta^{\left(i\right)}}=\theta^{\left(i\right)}-\alpha^{\left(i\right)}y^{\left(i\right)}x^{\left(i\right)}=0;\frac{\partial F}{\partial c}=\sum_{i=1}^{M}\alpha^{\left(i\right)}y^{\left(i\right)}=0.$$

As a result, we can describe weights as:(30)$$\theta =\sum_{i=1}^{M}\alpha^{\left(i\right)}y^{\left(i\right)}x^{\left(i\right)},$$
as well as the simultaneous issue as:(31)$$\underset{\alpha^{\left(i\right)}}{min}\left\{\frac{1}{2}\sum_{i,j}\alpha^{\left(i\right)}\alpha^{\left(j\right)}y^{\left(i\right)}y^{\left(j\right)}{\left(x^{\left(i\right)}\right)}^{T}x^{\left(j\right)}-\sum_{i=1}^{M}\alpha^{\left(i\right)}\right\},$$
assuming that:(32)$$\alpha^{\left(i\right)}\geq 0,$$
for every *i* in the training set i=1,…,M as well as:(33)$$\sum_{i=1}^{M}\alpha^{\left(i\right)}y^{\left(i\right)}=0.$$

It is possible to introduce nonlinearity into the optimization problem by expanding it to all kernel functions $K(x^{\left(i\right)},x^{\left(j\right)})$. This kernel approach is used instead of the dot product in the preceding dual problem:(34)$$\underset{\alpha^{\left(i\right)}}{min}\left\{\frac{1}{2}\sum_{i,j}\alpha^{\left(i\right)}\alpha^{\left(j\right)}y^{\left(i\right)}y^{\left(j\right)}K\left(x^{\left(i\right)},x^{\left(j\right)}\right)-\sum_{i=1}^{M}\alpha^{\left(i\right)}\right\}.$$

For instance, we may define the Gaussian kernel function by:(35)$$K\left(x^{\left(i\right)},x^{\left(j\right)}\right)=\exp\left(-\gamma ||x^{\left(i\right)}-x^{\left(j\right)}|{|}^{2}\right).$$

This necessitates more Euclidean distance computations. A detailed explanation of each stage of the algorithm is given below:1.kernel function and parameters initialization: Each parameter utilized by a kernel function must have its value initialized. Choose a relevant kernel function for the problem at hand and then generate the corresponding kernel matrix;2.Parameters and classical information represented by quantum states: In this stage, the objective function is segmented, and its components are recorded in qubits. Binary strings may be used to represent the conventional data:
(36)$$x\to b={\left(b_{1},b_{2},\ldots ,b_{m}\right)}^{T},$$
with $b_{i}=\left\{0,1\right\}$ for i=1,…,m. Then, the binary strings may be simply converted into *k* qubit quantum states:
(37)$$|b_{1},b_{2},\ldots ,b_{m}\rangle ,$$
which form a Hilbert space of $2^{k}$ dimensions covered by basis {|00…0〉,|10…0〉,…,|11…1〉};3.The quantum minimization subroutine examines objective function space: The Grover technique determines the optimal value of $\alpha^{\left(i\right)}$ that resolves for θ and *c* by searching the space of all possible objective functions. By first generating a superposition of all potential inputs to a quantum oracle *O* expressing the objective function, this procedure achieves a global minimum for the SVM optimization problem. This subroutine’s measurement yields the answer with a high degree of probability.

The Grover approach decreases the temporal complexity of the given SVM optimization problem from O(N) to $O(\sqrt{N})$, in which *N* is the training vector’s dimension, and provides a global minimum. The computation of a kernel matrix is one of the most time-consuming aspects of any SVM algorithm, with a computational complexity of $O(M^{2}N)$. The approach has the same limitations as that of the quantum subroutine GroverOptim, namely that the algorithm could not produce acceptable results due to quantum noise. It is assumed that the objective function of SVM is provided as an input to the algorithm by a quantum oracle. Another quantum technique for SVM, introduced in [4], demonstrates an unconstrained exponential speed-up, as we shall discuss below.

#### 3.3.2. Quantum Least Square SVM

The quantum variant of the Least Square SVM is a quantum machine learning breakthrough, in which Rebentrost et al. [4] obtained a significant exponentially speedup gain in the formulation of support vector machine. This algorithm transforms the optimization issue into such a linear equation system that includes the kernel matrix in handling the optimization model. By handling the least squares method and accelerating the kernel matrix computations, we obtain a significantly more efficient method for solving the linear equations system. We use quantum advantages in the following areas to improve the efficiency of the SVM:Quantum random access memory (QRAM) data translation: Preparing the collected data as input into the quantum device for computation is among the difficult tasks in QML. QRAM aids in transforming a collection of classical data into its quantum counterpart. QRAM requires $O(log_{2}\;d)$ steps to retrieve data from storage in order to reconstruct a state, with *d* representing the feature vector’s dimension;Computation of the kernel matrix: The kernel matrix is mostly determined by the examination of the dot product. Therefore, if we obtain speedup benefits of the dot product performance in the quantum approach, this will result in an overall speedup increase in the computation of the kernel matrix. With the use of quantum characteristics, the authors of [5] provide a quicker method for calculating dot products. A quantum paradigm is also used for the computation of the normalized kernel matrix inversion $K^{-1}$. As previously said, QRAM requires only $O(log_{2}\;d)$ steps to request for recreating a state, hence a simple dot product using QRAM requires $O(\epsilon^{-1}\;log_{2}\;d)$ steps, where ϵ represents a desired level of accuracy;Least square formulation: The quantum implementation of the exponentially faster eigenvector performance makes the speedup increase conceivable during the training phase in matrix inversion method and non-sparse density matrices [5].

The formulation for the least square SVM is as follows:(38)$$\hat{A}\left(|b,\vec{\alpha }\rangle \right)=|\vec{y}\rangle ;\hat{A}=\left[\begin{array}{cc} 0 & 1^{T}\\ 1 & K+\gamma^{-1}I \end{array}\right],$$
using $\hat{A}=\frac{A}{TrA}$ Normalized TrA, and TrA represents a trace of *A*. The aim will be solving the optimization issue $\hat{A}\left(|b,\vec{\alpha }\rangle \right)=|\vec{y}\rangle$, where *b* and $\vec{\alpha }$ are parameters, and *K* denotes the kernel matrix.

#### 3.3.3. Quantum Linear Regression

The aim of any regression issue is to make an accurate prediction of the continuous value of a target variable $y_{i}\in R$, based on a collection of input characteristics $x^{\left(1\right)},x^{\left(2\right)}\ldots x^{\left(N\right)}$, which may be expressed as an N-dimensional input data vector $x\in R^{N}$. It is assumed in linear regression that the output is a linear composition of the input characteristics and an unavoidable error factor $\epsilon_{i}$, as shown below:(39)$$y_{i}=\theta_{1}x^{\left(1\right)}+\theta_{2}x^{\left(2\right)}+\ldots +\theta_{N}x^{\left(N\right)}+b+\epsilon_{i}=\sum_{i=1}^{N}\theta_{i}x\left(i\right)+b+\epsilon_{i}.$$

The parameters of the model we would like to learn are the $\theta_{i}$ values relating to each feature and the intercept *b*. If the parameters $\theta_{i}$, where i∈{1,2,…N} are represented with such a vector θ∈R, the linear connection from Equation (Equation 39) may therefore be simplified as follows:(40)$$y_{i}/x_{i}=\theta^{T}x_{i}+b+\epsilon_{i}.$$

The phrase $y_{i}/x_{i}$i represents the value of $y_{i}$ as a function of $x_{i}$. Therefore, $\epsilon_{i}$ would be the fundamental component which has zero association with the input characteristics, and, as a result, cannot be learned. Nevertheless, given $x_{i}$, we can precisely define the phrase $\theta^{T}x_{i}+b$. Assuming that the error $\epsilon_{i}$ has a normal distribution having zero mean and limited standard deviation $\sigma_{i}$, the following expression may be written:(41)$$\epsilon_{i}∼N\left(0,\sigma^{2}\right),$$
and $\theta^{T}x_{i}+b$ was indeed constant based on the value within the feature vector $x_{i}$; hence, we may affirm:(42)$$\theta^{T}x_{i}+b+\epsilon_{i}∼N\left(\theta^{T}x_{i}+b,\sigma^{2}\right),$$
$$\to y_{i}/x_{i}∼N\left(\theta^{T}x_{i}+b,\sigma^{2}\right).$$

Given the input feature, the target label $y_{i}$ follows a normally distributed with mean $\theta^{T}x_{i}+b$ and standard deviation σ. As can be seen below, in linear regression, the conditional mean of the distribution is used to make predictions:(43)$${\hat{y}}_{i}=E\left[y_{i}/x_{i}\right]=\theta^{T}x_{i}+b.$$

Minimizing the square of the error term $\epsilon_{i}$ at every data point enables the identification of the model parameters θ and *b*. The bias factor *b* may be used as a parameter through the parameter vector θ associated with the constant feature of 1 for convenience of notation. This results in the prediction ${\hat{y}}_{i}=\theta^{T}x_{i}$ where θ and $x_{i}$ represent N+1-dimensional vectors. Through this reduction in complexity, the equation system describing the *M* data points may be expressed in matrix notation below:(44)$$\left[\begin{array}{c} x_{1}^{T}\\ x_{2}^{T}\\ \ldots \\ x_{i}^{T}\\ \ldots \\ x_{M}^{T} \end{array}\right]\theta =\left[\begin{array}{c} {\hat{y}}_{1}\\ {\hat{y}}_{1}\\ \ldots \\ {\hat{y}}_{i}\\ \ldots \\ {\hat{y}}_{M} \end{array}\right].$$

Equation (Equation 44) may be rewritten as follows using a matrix representation of the input feature vectors $X\in R^{M\times (N+1)}$ and a vector representation of the prediction ${\hat{Y}}_{i}\in R^{M}$:(45)$$X\theta =\hat{Y}.$$

Therefore, if we let the vector $Y\in R^{M}$ represent the actual targets $y_{i}$ in all *M* sets of data, we can calculate the error vector ϵ∈R as below:(46)$$\epsilon =Y-\hat{Y}=Y-X\theta .$$ The loss function may be expressed also as the average of the squared errors in the predictions over all points in the data:(47)$$L\left(\theta \right)=\frac{1}{M}\sum_{i=1}^{M}\epsilon_{i}^{2}=\frac{1}{M}\epsilon^{T}\epsilon =\frac{1}{M}{\left(Y-X\theta \right)}^{T}\left(Y-X\theta \right).$$

The parameter may be determined by minimizing the loss L(θ) for a range of values of θ. The minima may be found by setting the gradient of the loss L(θ) relative to θ to zero vector, as illustrated below:(48)$$\nabla_{\theta }=-\frac{2}{M}X^{T}\left(Y-X\theta \right)=0;$$
then,
$$\left(X^{T}X\right)\theta =X^{T}Y.$$ The matrix $\left(X^{T}X\right)$ may be thought of as the Hamiltonian of a quantum system due to its Hermitian character. Using the HHL technique, we may solve the matrix inversion issue posed by Equation (Equation 48) in order to obtain the model parameter θ.

#### 3.3.4. Quantum K-Means Clustering

Grover’s search algorithm and the Euclidean distance computation approach may be used to provide the quantum equivalent of conventional k-means clustering. The processes are identical to those of the traditional *k*-means algorithm, only with the exception that quantum routines rather than conventional ones are used to execute each step. The steps are detailed below.

1.Initialization: Set the *k* cluster $u_{1},u_{2},\ldots ,u_{k}\in R^{n}$ by using heuristic comparable to that of the traditional *k*-means algorithm. For instance, *k* data points may be randomly selected as the first clusters;2.Until Convergence:(a)In every piece of data, $x_{i}\in R^{n}$ defined with its magnitude $\left|\right|{\vec{x}}_{i}{\left|\right|}_{2}$ saved conventionally as well as the unit norm $|x_{i}\rangle$ saved like a quantum state, the distance is computed by using quantum Euclidean distance computation procedure with every one of the *k* cluster centroids:
(49)$$d\left(i,j\right)=\left|\right|\left(x_{i}-u_{j}\right){\left|\right|}^{2}=4Z\left(P\left(|0\rangle \right)-0.5\right),$$
with j∈{1,2,…k};(b)Apply the search technique developed by Grover to allocate every $x_{i}$ in the data set to a single *k* clusters. As seen below, the oracle implemented in Grover’s search method must be capable of taking the distance d(i,j) and then allocate the proper cluster $c_{i}$ according to the formula below:
(50)$$c_{i}=argmin_{j}\left|\right|\left(x_{i}-u_{j}\right){\left|\right|}^{2}c_{i}\in \left\{1,2,\ldots k\right\};$$(c)The mean of each cluster is determined after assigning each piece of data $x_{i}$ to its cluster $c_{i}\in \left\{1,2,\ldots k\right\}$:
(51)$$u_{j}=\frac{1}{N_{j}}\sum_{c_{i}=j}x_{i},$$
where $N_{j}$ represents the total number of data that might be assigned to cluster *j*.

The method converges when the clusters of data points no longer change with each iteration. Conventional *k*-means clustering does have an iteration complexity equal to O(MNk)). For each data point, the computational complexity of the distance to a cluster is O(N), wherein *N* represents the number of features. The complexity is O(Nk) for each data point, as there are *k* clusters. Moreover, the total complexity for every iteration is O(MNk), as the complexity for each of the *M* points of data are O(Nk). The advantage of the quantum k-means implementation is that the complexity of computing the quantum Euclidean distance between every data point in a cluster is just Mlog(N)k for a high value of a feature size *N*. Notably, for conventional and quantum k-means, the complications of allocating every data point towards the right cluster using distance minimization are not taken into account. In this aspect, if built appropriately, Grover’s technique, which is used to allocate data points to its respective clusters, may give an additional speedup.

#### 3.3.5. Quantum Principal Component Analysis

The classical Principal component analysis (PCA) is a crucial dimensionality reduction approach. We might claim that PCA works by transposing the covariance matrix of the data, $C=\sum_{j}\vec{u}{\vec{u}}_{j}^{T}$, onto its diagonal. The covariance matrix summarizes how the various data elements are related to one another. To write the covariance matrix in terms of its eigenvectors and eigenvalues, we write $C=\sum_{j}e_{k}{\vec{v}}_{k}{\vec{v}}_{k}^{†}$, wherein $e_{k}$ represents the eigenvalues corresponding to the eigenvectors $v_{k}$. Principal components refer to the few eigenvalues that are relatively big compared to the rest. Therefore, each of the primary components is regarded as a new feature vector.

The traditional PCA technique has a runtime complexity of $O(d^{2})$, wherein *d* corresponds to the size of the Hilbert space. For quantum PCA [14], a quantum state is transferred into a randomly selected data vector using QRAM. $\rho =\frac{1}{N}\sum_{j}|u_{j}\rangle \langle u_{j}|$ is the density matrix of the resultant state, with *N* being the cardinality of the input vector collection. Applying density matrix exponentiation in conjunction with the continuous sampling of the data and quantum phase estimation approach, we may deconstruct the input vectors into their principal components. The quantum PCA method has a runtime complexity of $O({\left(logd\right)}^{2})$.

## 4. ML vs. QML Benchmarks

In this section, we perform the Variational Quantum Classifier and compare it with several classical classifiers; next, we implement QSVM and compare it with the classical SVM, and then the implementation of the Quantum Convolutional Neural Network is discussed in the last sub-section.

### 4.1. Variational Quantum Classifier

The Variational Quantum Classifier (VQC) is based on the use of a function f(x,θ)=y which can be implemented in a quantum computer using a circuit denoted as $S_{x}$ in order to encode the input data *x* toward a quantum state and especially as amplitudes of the state, then a quantum circuit denoted as $U_{\theta }$, finally a one simple qubit measurement. This latter measurement yields the likelihood of the VQC predicting ’0’ or ’1’, which may be used to determine the prediction of a binary classification. The circuit parameters (θ) of this classification are also trainable, and a variational technique may be employed to train these parameters [77]. The four steps of VQC are shown in Figure 14 and explained below.

State preparation: To be able to encode classical data into quantum states, we use particular operations to help us work with data in a quantum circuit. As mentioned earlier, quantum encoding is one these methods that consists of representing classical data in the form of a quantum state in Hilbert space employing a quantum feature map. Recall that a feature map is a mathematical mapping that allows us to integrate our data into higher dimensional spaces, such as quantum states in our case. It is similar to a variational circuit in which the parameters are determined by the input data. It is essential to emphasize that a variational circuit depends on parameters that can be optimized using classical methods;The model circuit: The next step is the model circuit, or the classifier in precise terms. $\varphi^{\prime }$ is generated using a parameterized unitary operator $U_{\theta }$ applied to the feature vector noted as φ(x) which became a vector of a quantum state in an n-qubit system (in the Hilbert space ). The model uses a circuit that is composed of gates which change the state of the input and are built on unitary processes, and they depend on external factors that can be adjusted. $U_{\theta }$ translates φ(x) into another vector $\varphi^{\prime }$ with a prepared state |φ(x)〉 in the model circuit. $U_{\theta }$ is comprised of a series of unitary gates;Measurement: We take measurements in order to obtain information from a quantum system. Although a quantum system has an infinite number of potential states, we can only recover a limited amount of information from a quantum measurement. Notice that the number of qubits is equal to the amount of results;Post-process: Finally, the results were post-processed including a learnable bias parameter and a step function to translate the result to the outcome 0 or 1.

#### 4.1.1. Implementation

The architecture that we have implemented is inspired from [77], as we have already mentioned. The goal is to encode the real vectors as amplitude vectors (amplitude encoding) and training the model to detect the first two classes of the Iris dataset.

Dataset: Our classification data sets made up of three sorts of irises (Setosa, Versicolour, and Virginica), and containing four features that are Sepal Length, Sepal Width, Petal Length and Petal Width. In our implementation, we used the two first classes:Implemented models:-VQC: The Variational Quantum Classifiers commonly define a “layer”, whose fundamental circuit design is replicated to create the variational circuit. Our circuit layer is composed of a random rotation upon each qubit, and also a CNOT gate that entangles each qubit with its neighbor. The classical data were encoded to amplitude vectors via amplitude encoding;-SVC: A support vector classifier (SVC) implemented by the sklearn Python library;-Decision Tree: Is a non-parametric learning algorithm which anticipates the target variable through learning decision rules;-Naive Bayes: Naive Bayes classifiers utilize Bayes theory with the assumption of conditional independence in between each pair of features;Experimental Environment: We use the Jupyter Notebook and PennyLane [80] (A cross-platform Python framework for discrete programming of quantum computing) for developing all the codes and executing them on IBM Quantum simulators [81]. We implemented three classical Scikitlearn [82] algorithms to accomplish the same classification task on a conventional computer in order to compare their performance with that of the VQC. We also used the end value of the cost function and the test accuracy as metrics for evaluating the implemented algorithms.

#### 4.1.2. Results

As it can be plainly observed from the Table 4, our VQC works slightly better than the best result obtained with the SVC in both terms of accuracy and cost function value. The probability of a model guessing the correct class in the classical example is less than 0.96, but the probability of the quantum classifier is 1 with the value of the cost function equal to 0.23, which shows that the quantum classifier made small errors in the data compared to SVC and Decision Tree.

One may assume that the effort required to employ quantum algorithms is exaggerated. However, we have to look at the whole process, in which we reduced the dimension of the data set in order to handle it by only two qubits in order to use a smaller circuit depth to prepare quantum states. Despite this, the results are promising.

### 4.2. SVM vs. QSVM

Quantum Support Vector Machine (QSVM) is a quantum variant of the standard SVM algorithm that uses quantum principles to perform computations. QSVM employs the advantages of quantum hardware and quantum software to enhance the performance of classic SVM algorithms that perform on classical computers with GPUs or CPUs. The initial procedure QSVM is to encode the conventional data which contains specific categories (class, features, dimension, etc.) giving the information about the data we wish to classify. Current quantum hardware functions are yet to achieve full capability as there is a limited number of qubits accessible. Therefore, we need to decrease the features of the data to render it compatible with the available qubits. The principal component analysis (PCA) is typically employed in machine learning for this kind of procedure. Next, the classical data are required to be mapped towards quantum input in order to be treated by the quantum computer. These mapped data are called a feature map which operates as a quantum feature for classical data. Finally, the proper machine learning algorithm needs to implemented to attain the best classification result. This implementation covers many processes including dividing the data into training and testing sets, determining the amount of qubits to be employed, defining the classes of data. Additionally, we need to define the algorithm as well as parameters for how many iterations this should be carried out with the depth of the circuit.

#### 4.2.1. Implementation

The objective of this implementation is to compare the performances of the classical SVM and the QSVM which is implemented on a dataset that represents the real world using a quantum computer.

Breast cancer dataset: The Wisconsin Diagnostic Breast Cancer dataset (WDBC) of UCI machine learning repository is a classification dataset that includes breast cancer case metrics. There are two categories: benign and malignant. This dataset contains information on 31 characteristics that define a tumor, among which are: average radius, mean perimeter, mean texture, etc., and a total of 569 records and 31 features;Principal Component Analysis: The existing quantum computers are yet to attain full potential as their functions are constrained by the limited number of accessible qubits, noise, and decoherence. Thereby, a dataset with exceptionally big dimensionality is difficult to be implemented. In addition, this is where Principal Component Analysis (PCA) comes to help. As mentioned above, PCA is the technique that reduces the huge dimension into smaller-scaled dimensions keeping the correlation given in the dataset. The principal components are orthogonal since they are the eigenvectors of the covariance matrix. Generally, we can assume that the data summarized can be processed by a quantum computer. Using PCA offers us the flexibility to leave out some of the components without losing much information and therefore minimizing the complexity of the problem as illustrated in Figure 15, where we reduced our dataset in order to handle it with a quantum computer;Quantum Feature Map: As we discussed earlier, the Kernel techniques are the set of algorithms for pattern analysis or recognition of the data points. Kernel techniques map data to higher dimensional spaces in order to ease data structure and separability. The data may be translated not only to higher-level space but to an endless dimension. Kernel “trick” is the technique of replacing the inner products of two vectors in our algorithm with the kernel function. In other hand, we have Quantum Kernel techniques which is a method for identifying a hyperplane that is performing a nonlinear transformation to the data, This is termed as a “feature map”. We may either apply available Qiskit (an open-source quantum computing framework created by IBM.) feature maps including ZZ feature map, Z feature map, or a Pauli feature map that has Pauli (X, Y, Z) gates, or create a custom feature map based on the dataset compatibility. The quantum feature map can be built by using Hadamard gates with entangling unitary gates between them. In our case, we used the ZZ feature map.The needed number of qubits is proportional to the data dimensionality. By altering the angle of unitary gates to a certain value, data are encoded. QSVM uses a Quantum processor to estimate the kernel in the feature space. During the training step, the kernel is estimated or computed, and support vectors are obtained. However, in applying the Quantum Kernel method directly, we should encode our training data and test data into proper quantum states. Using Amplitude Encoding, we may encode the training samples in a superposition inside one register, while test examples can be encoded in a second register.

#### 4.2.2. Results

We have implemented SVM on the breast cancer dataset from the Scikit learn library in a classical computer using SVM as well as using QSVM on a quantum simulator in the IBM cloud using its quantum machine learning library Qiskit [81], in order to compare their performances.

As we can observe from Figure 16, the QSVM has predicted accurately the classes of our test data, and we obtained an accuracy around 90%. Notice that, on the diagonal of the kernel matrix, we have only black squares, which means that each point is at a distance of zero from itself. In addition, what we are seeing in the kernel matrix in Figure 16 and Figure 17, is the calculated distance of the kernel in the space with higher dimensions. Thus, we may say that, with QSVM, our dataset can simply classify, due to the quantum feature map method which encodes data in a space with a higher dimension.

### 4.3. CNN vs. QCNN

Convolutional Neural Networks (CNNs) are a prominent model in computer vision that has gained a ton of potential for different machine learning tasks notably in the area of image recognition. The capacity of such networks to retrieve features from input in a typical hierarchical structure provides the majority of their advantages. Several transformation layers, mainly the convolutional layer that provides its name to the model, are employed to retrieve these features. To perform the CNN model on a quantum computer, we implement another kind of transformation layer named quanvolutional layer or quantum convolution utilized in Quanvolutional Neural Networks model (QNNs), which is introduced by Henderson et. al. [29]. Quanvolutional layers perform on data input by encoding it through a series of arbitrary quantum circuits, in a manner that is equivalent to the task of the aleatory convolutional filter layers. In this subsection, we briefly review the architectural design of Convolutional Neural Networks (CNNs) and then the architectural design of Quanvolutional Neural Networks (QNNs) that are also known as Quantum Convolutional Neural Networks (QCNNs). Then, we evaluated the performance advantage of quantum transformations by making a comparison between the CNN and QNN models based on the MNIST dataset.

Convolutional Neural NetworksConvolutional neural networks (CNNs) are a specialized sort of neural networks, built primarily for time series or image processing. They become presently the most frequently used models for image recognition applications. Their abilities have indeed been applied in several sectors such as gravitational wave detection or autonomous vision. Despite these advances, CNNs struggle from a computational limitation which renders deep CNNs extremely pricey in reality. As Figure 18 illustrates, a convolutional neural network generally consists of three components even though the architectural implementation varies considerably:1.Input: The most popular input is an image, although significant work has also been carried out on so-called 3D convolutional neural networks which can handle either volumetric data (three spatial dimensions) or videos (two spatial dimensions + one temporal dimension). For the majority of the implementations, the input needs to be adjusted to correspond to the specifics of the CNN used. These include cropping, lowering the size of the image, identifying a particular region of interest, and also normalizing pixel values to specified regions. Images, and even more widely layers of a network, may be represented as tensors. The tensor is a generalization of a matrix into extra dimensionality. For example, an image of a height *H* and a width *W* may indeed be visualized as just a matrix in $R^{H\times W}$, wherein every pixel represents a greyscale ranging between 0 and 255. Furthermore, all three channels in color RGB (Red Green Blue) should be taken into account, simply layering 3 times the matrix for every color. A whole image is thus viewed as a three-dimensional vector in $R^{H\times W\times D}$ wherein *D* represents the number of channels;2.Feature Learning: The feature learning is built of three principal procedures, executed and repeated in any sequence: Convolution Layers, usually followed with an Activation Function and Pooling Layers at the end of this feature learning process. We indicate by *l* the present layer.-Convolution Layer: Each *l*th layer is combined by a collection filter named kernels. The result of this procedure would be the (l+1)th layer. The convolution using a simple kernel may be considered to be like a feature detector, which will screen through all sections of the input. If a feature described by a kernel, for example, a vertical border, is present through some area of the input, it will be highly valuable at the corresponding point of the outcome. The outcome is known as the feature map of the whole convolution;-Function: Just like in normal neural networks, we add certain nonlinearities also named activation functions. These functions are necessary for a neural network in order to be capable of learning any function. As in implementation of a CNN, every convolution is frequently followed by a Relu (Rectified Linear Unit function). This is a basic function that sets all negative numbers of the output to zero, then leaves the positive values as they are;-Pooling Layer: The downsampling method that reduces the dimensions of the layer, particularly optimizing the calculation. Furthermore, it provides the CNN the capability to learn a form invariant to lower translations. Almost all of the cases, either we use a Maximum Pooling or perhaps an Average Pooling. Maximum Pooling consists of swapping a subsection of P×P components just by another with the biggest value. Average Pooling achieves this by averaging all numbers. Note that the number of a pixel relates to how often a certain feature is represented in the preceding convolution layer;3.Classification/Fully Connected Layer: Following a specific set of convolution layers, the inputs had been properly handled such that we may deploy a fully connected network. The weights link every input to every output, wherein inputs are all components of the preceding layer. The final layer must include a single node for each possible label. The node value may be read as the probability that the input image belongs to the specified class.Quanvolutional Neural NetworksThe Quanvolutional Neural Networks (QNNs) are essentially a variant of classical convolutional neural networks with an extra transformation layer known as the quanvolutional layer, or quantum convolution that is composed of quanvolutional filters [29]. When applied, the latter filters to a data input tensor; then, this will individually generate a feature map by changing spatially local subsections of this input. However, unlike the basic handling data element by element in matrix multiplication performed by a traditional convolutional filter, a quanvolutional filter transforms input data using a quantum circuit that may be structured or arbitrary. In our implementation, we employ random quantum circuits as proposed in [29] for quanvolutional filters rather than circuits with a specific form, for simplicity and to create a baseline.This technique for converting classical data using quanvolutional filters can be formalized as follows and illustrated in Figure 19:1.Let us simply begin with a basic quanvolutional filter. The latter filter employs the arbitrary quantum circuit *q* that accepts as inputs subsections of images of dataset a *u*. Each input is denoted by the letter $u_{x}$, with each $u_{x}$ being a two-dimensional matrix of length *n*-by-*n*, where *n* is greater than 1;2.Despite the fact that there are a variety of methods for “encoding” $u_{x}$ as an initial state of *q*, we chose one encoding method *e* for every quanvolutional filter, and we define the resulting state of this encoding with $i_{x}=e\left(u_{x}\right)$;3.Thereafter, the application of the quantum circuit *q* to the beginning state $i_{x}$, and the outcome of the quantum computing was indeed the quantum state $o_{x}$, as defined by the relation $o_{x}=q\left(i_{x}\right)$ that also equals $q(e\left(u_{x}\right))$;4.In order to decode the quantum state $o_{x}$, we use *d* which is our decoding method that converts the quantum output into classical output using a set of measurements, which guarantees that the output of the quanvolutional filter is equivalent to the output of a simple classical convolution. The decoded state is described by $f_{x}=d\left(o_{x}\right)$ with $d(q\left(e\left(u_{x}\right)\right))$ in which $f_{x}$ represents a scale value;5.Lastly, we denote the full transformation achieved by the ’quanvolutional filter transformation’ [29] by *Q* of a data point $u_{x}$, which is also described as $f_{x}=Q\left(u_{x},e,q,d\right)$. Figure 19B represents one basic quanvolutional filter, showing the encoding, the applying circuit and the decoding process.

**Figure 18 entropy-25-00287-f018:**
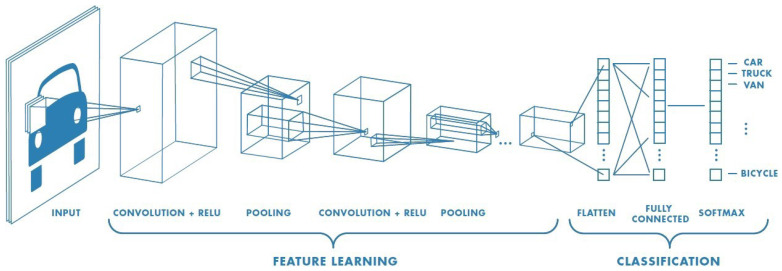
Representation of a CNN’s architecture. Source: Mathworks.

**Figure 19 entropy-25-00287-f019:**
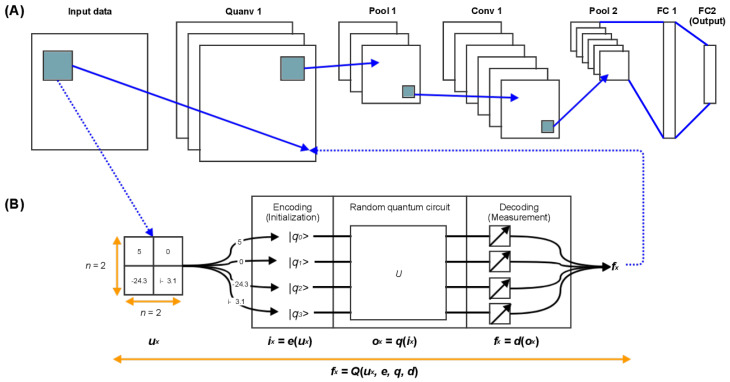
(**A**) represents a quanvolutional layer inside a network architecture. The quanvolutional layer has a series of quanvolutional filters (three in this case) that transform the entered data to various outcome feature maps. (**B**) represents the transformation of conventional data input and output of an arbitrary quantum circuit within the quanvolutional filter [29].

#### 4.3.1. Implementation

The aim of this implementation is to demonstrate the practicality of quanvolutional layers included in a classical CNN model, the feasibility to employ this quanvolutional technique using actual world data sets, and the possibility of using features created by the quanvolutional transformation.

Dataset: We used a subset of the MNIST (Modified or Mixed National Institute of Standards and Technology) dataset, which includes 70,000 handwritten 28 by 28 pixels in grayscale;Implemented models:-The CNN model: We used a simple model: Fully-connected layer containing ten output nodes and then a final activation function softmax in a pure classical convolutional neural network;-The QNN model: A CNN with one quanvolutional layer seems to be the simplest basic quanvolutional neural network. The single quantum layer will be the first, and then the rest of the model is identical to the traditional CNN model;Experimental Environment: The Pennylane library [80] was used to create the implementation, which was then run on the IBM quantum computer simulator QASM;Quanvolutional layer generation process described as follows:-A quantum circuit is created by including some small section of the input image, in our case a square of 2×2. A rotation encoding layer denoted by $R_{y}$ (with angles scaled by a factor π);-The system performs a quantum computation associated with a unit U. A variational quantum circuit or, quite generally, a randomized circuit could generate this unit;-The quantum system is measured using a computational basis measurement, which gives a set of four classical expectation values. The outcomes of the measurements can eventually be post-processed in a classical way;-Each value is translated into a distinct channels of one output pixel, as in a classical convolution layer;-In repeating the technique on different parts, the whole input image can be scanned, thus obtaining an output object structured as a multi-channel;Other quantum or classical layers can be added after the quanvolutional layer.

#### 4.3.2. Results

Before analyzing the performance of the QNN model and traditional CNN results, we should first confirm that the model works as planned; the addition of the quanvolutional layer to the model has resulted in a highest accuracy, and the more the training iterations there are, the more the accurate the model becomes. The value of the loss function in the QNN models is quite low, which shows that our model has fewer errors during testing. As shown in Table 5 below, the QNN has a slightly higher accuracy than the CNN which indicates that our QNN model has well predicted our test data which are not used in the training of the model. We can conclude that the generated quantum features were effective in constructing features for the MNIST data set for classification purposes.

In Figure 20, we clearly observe the reduction of the sampling of the resolution and certain local deformations generated by our quanvolutional layer. In addition, the general shape of the input image is preserved, as expected for a convolution layer.

Figure 21 displays the outcomes of this implementation. Our QNN model typically provides superior learning performance compared to the CNN model. The presented results prove that the quantum convolution layer might boost learning performance.

## 5. General Discussion

As shown in the results above, the three quantum algorithms outperformed their classical counterparts in terms of accuracy and cost function value. However, it is indeed challenging to affirm that a QML algorithm will ever be better compared to a classical machine learning algorithm through practice since a quantum computer is evolved in noisy intermediate-scale quantum (NISQ) and small-scale quantum devices, which inhibits us to use a limited amount of qubits and few features, and huge data processing on quantum devices is impracticable due to the loss of a large amount of important information. QML has a number of challenges that are required to be solved on both the hardware and software levels. Quantum computers with a high-qubit-count and interface devices, such as quantum random access memory (qRAM), which enables classical information to be recorded in quantum states structure, represents the biggest and most significant challenges. Small scale quantum computers with 50–150 qubits are commonly accessible through quantum cloud computing, and their hardware is distinct; we can observe ion-trapped devices (IonQ) and both superconducting devices (IBM or Rigetti). Additionally, there are devices consisting of topological qubits, NV center (diamond), photons, quantum dots and neutral atoms. When it comes to physically implementing the quantum devices, each company decides one or more technologies. Each has benefits and drawbacks, and there is no agreement or dominance of one in the literature. However, due to the size of these devices, the complexity of their implementation, and their ongoing development, their everyday access is not yet feasible. An upgrade in quantum hardware will enable an improvement in the performance of applied algorithms. Because the the potential number of output labels is proportional to the amount of qubits in the device, most of the used algorithms, such as image classification using QCNN, are constrained by the number of qubits. We may also find a vast selection of quantum simulators on many platforms, which enable the testing of quantum computers as if they were real and even imitate their noise.

## 6. Conclusions and Perspectives

This article reviews a selection of the most recent research on QML algorithms for a variety of issues which are more accurate and effective than classical algorithms in classical computing, especially focusing on the quantum variant of certain supervised machine learning algorithms that rely on the VQC. In addition, we discuss numerous ways for mapping data into quantum computers. We covered approaches of quantum machine learning such as quantum sub-processes. In addition, then, we compared the performance of various QML algorithms such as QSVM, VQC, and QNN with that of its classical counterparts.

The QSVM algorithm exceeded the classical SVM in performance and speed up, but when it performed on a small subset of the dataset. Most quantum machine learning techniques are at some point restricted by the absence of an appropriate quantum RAM (QRAM) which facilitates the mapping of classical input to a quantum computer. The number of practical machine learning algorithms in the area of QML is expected to rise as QRAM implementations reach a stable state.

The VQC obtains a greater advantage over certain classical classifiers when implemented on a quantum computer with a smaller circuit depth, employing only two qubits to encode a subset of the dataset. Furthermore, we aim for a similar result on a large dataset with the existence of large number of qubits.

The result produced by the QNN demonstrated that the model can be a solution in several classification tasks, and it is a more efficient and effective learning model when employed with a classical CNN model. Moreover, the QNN model may also predict highly efficient results in much more sophisticated and large-scale training using quantum computers of the NISQ (Noisy Intermediate-Scale Quantum) era.

In conclusion, comprehensive benchmarks on large datasets are needed to systematically examine the influence of growing input data. In addition, in the near-term, the quantum encoding (feature maps) approach is a strong and theoretically interesting approach to think about in order to successfully implement quantum machine learning algorithms.

As a future work, we are currently developing new quantum machine learning algorithms that are able to handle a large actual world datasets which can be handled on currently quantum computers. Our next objective is to examine the implementation of classical machine learning models on quantum computers.

## Figures and Tables

**Figure 1 entropy-25-00287-f001:**
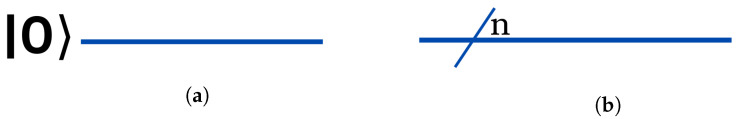
Representations of wires of quantum circuit. (**a**) Single quantum circuit wire; (**b**) Quantum circuit wire with n qubits.

**Figure 2 entropy-25-00287-f002:**
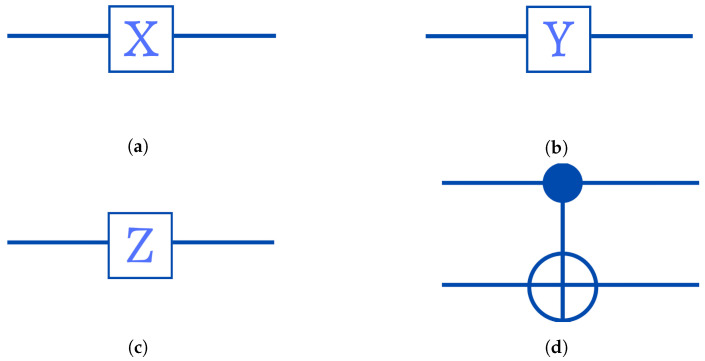
Circuit representations of the four most used gates in quantum circuits. (**a**) The Pauli-Y gate; (**b**) The Pauli-Y gate; (**c**) The Pauli-Z gate; (**d**) The CNOT gate.

**Figure 3 entropy-25-00287-f003:**
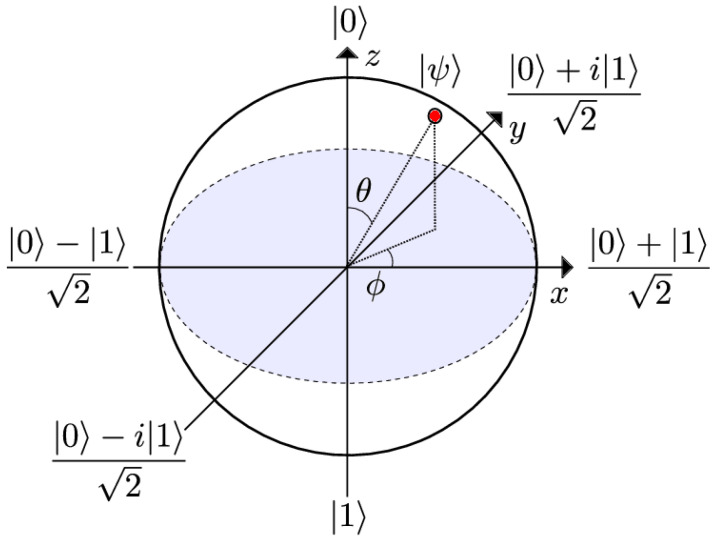
Bloch sphere representation of the state of qubit. Reprinted with permission from Ref. [37]. 2014, Anton Frisk Kockum.

**Figure 4 entropy-25-00287-f004:**
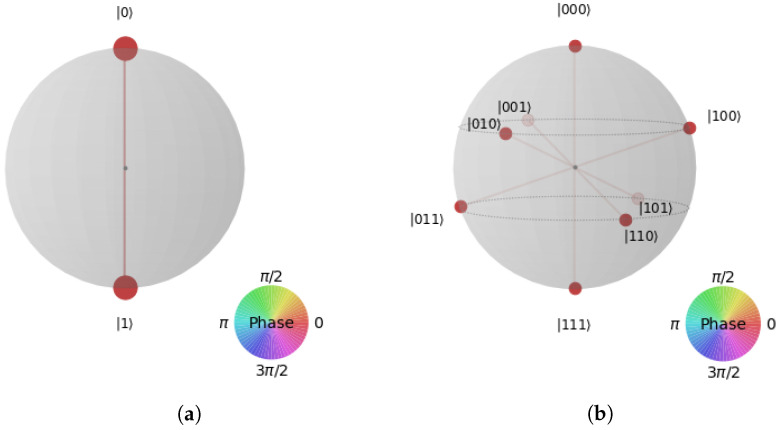
Representations of single qubit and multi-qubit in Q-sphere. (**a**) Representation of a superposition state; (**b**) Representation of three-qubit states.

**Figure 5 entropy-25-00287-f005:**
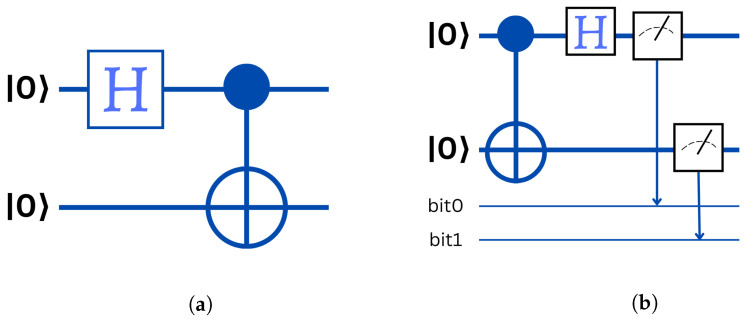
Quantum circuits of Bell state and Bell measurement. (**a**) Bell state; (**b**) Bell measurement.

**Figure 6 entropy-25-00287-f006:**
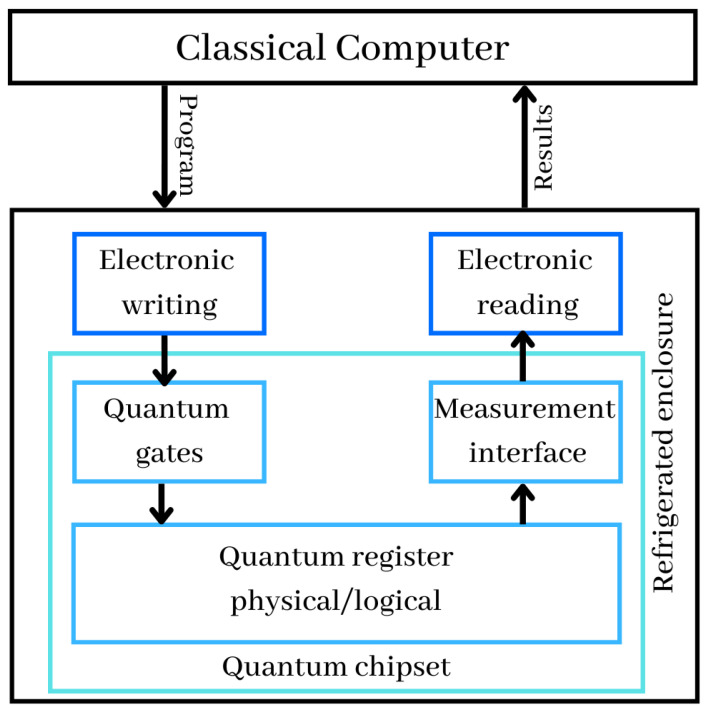
The quantum computer’s architecture.

**Figure 7 entropy-25-00287-f007:**
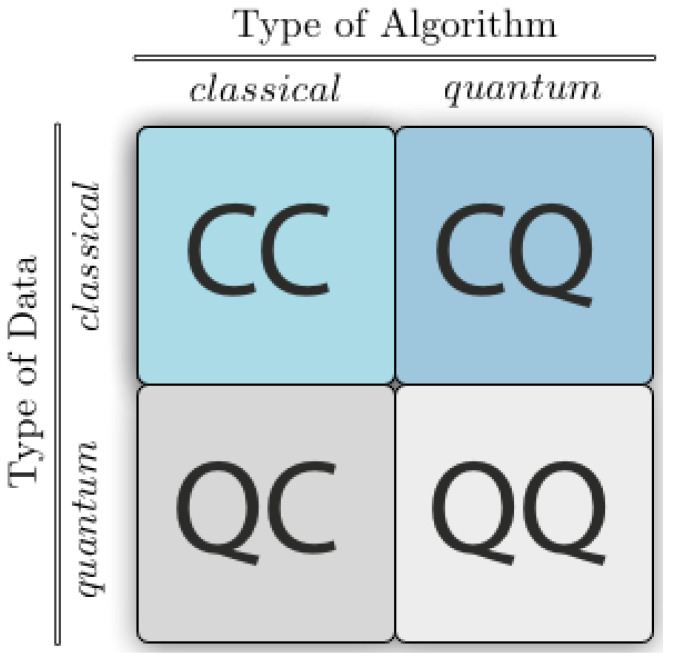
Four different ways to combine quantum computing with machine learning. Reprinted with permission from Ref. [47]. 2017, Maria Schuld.

**Figure 8 entropy-25-00287-f008:**
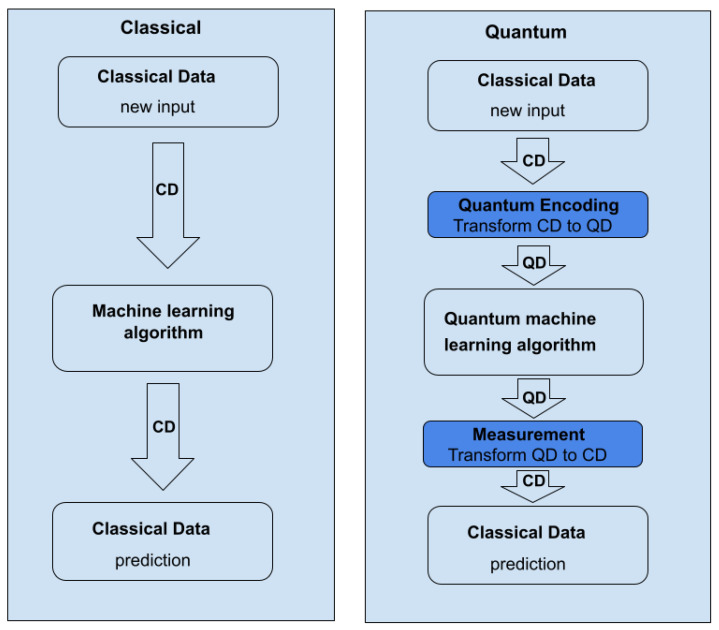
Processing techniques of conventional machine learning and quantum machine learning. CD represents classical data and QD represents quantum data.

**Figure 9 entropy-25-00287-f009:**
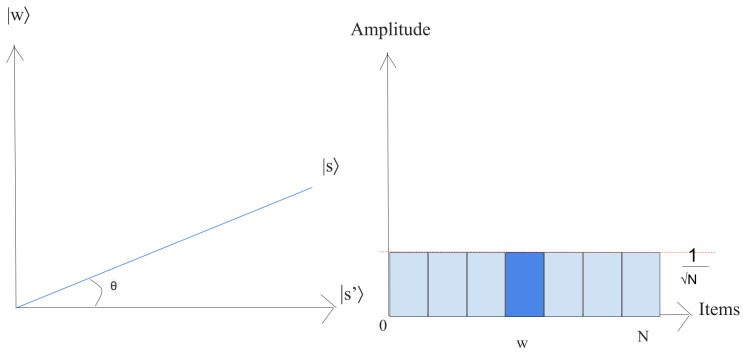
Geometric visualization and the condition of the amplitude of the state |s〉.

**Figure 10 entropy-25-00287-f010:**
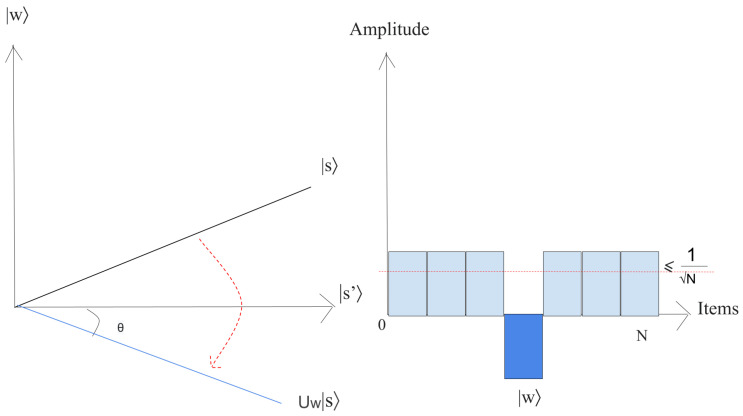
Geometric visualization and the condition of the amplitude after the implementation of the $U_{w}$ operator.

**Figure 11 entropy-25-00287-f011:**
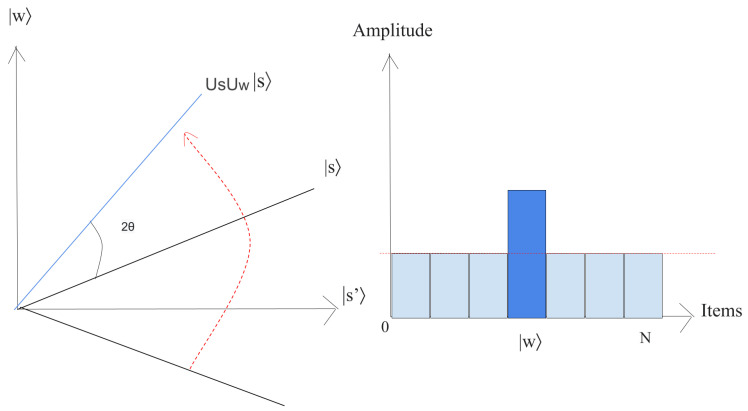
Geometric visualization and the condition of the amplitude after the implementation of the $U_{s}$ operator.

**Figure 12 entropy-25-00287-f012:**
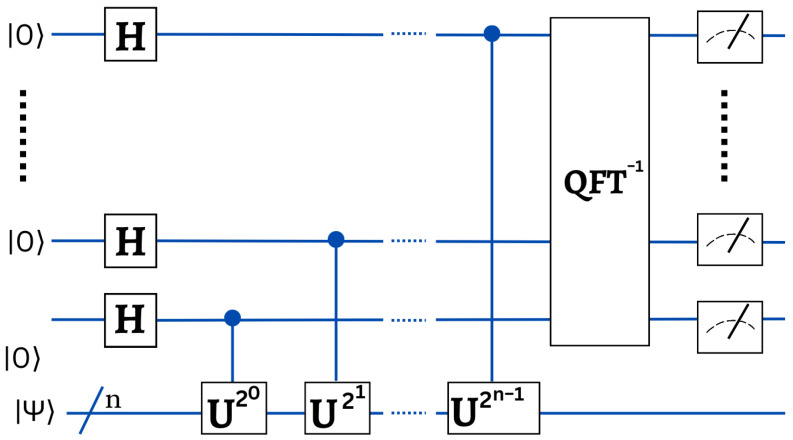
Quantum phase estimation circuit. $QFT^{-1}$ is just the inverse QFT.

**Figure 13 entropy-25-00287-f013:**
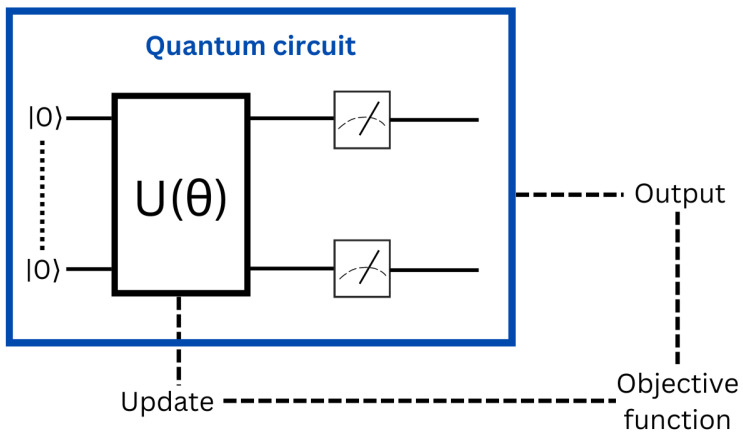
Representation of a variational quantum circuit optimization scheme.

**Figure 14 entropy-25-00287-f014:**
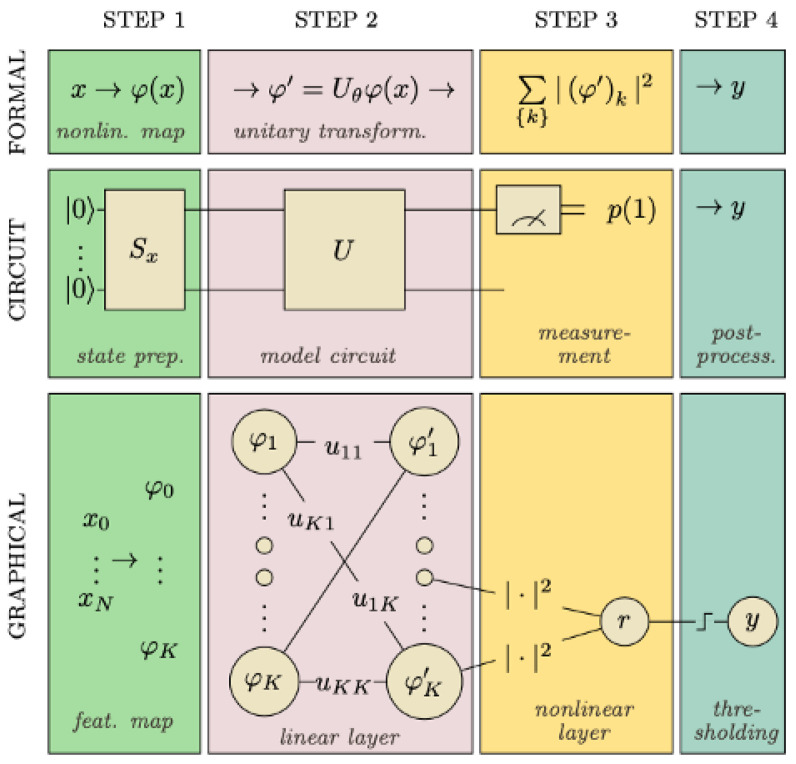
The classification with the quantum classifier consists of four phases, each represented by a distinct color, and can be examined from three different perspectives: a formal mathematical diagram, a quantum circuit diagram, and a graphical neural network diagram [77].

**Figure 15 entropy-25-00287-f015:**
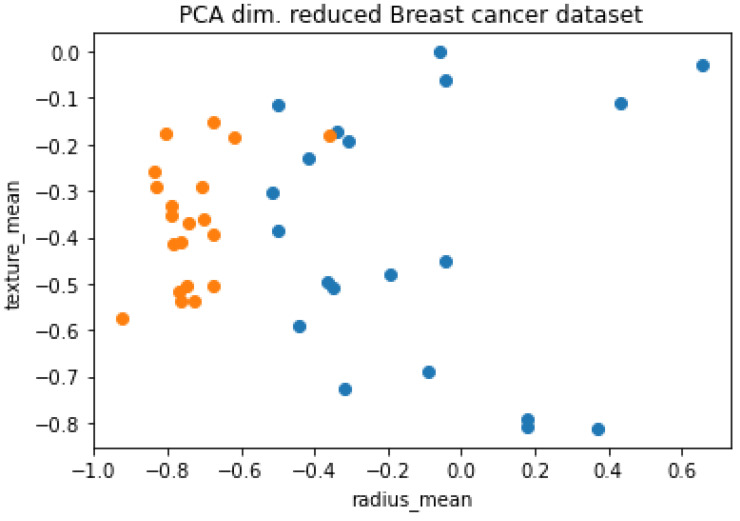
The set of data (20 total) after performing PCA.

**Figure 16 entropy-25-00287-f016:**
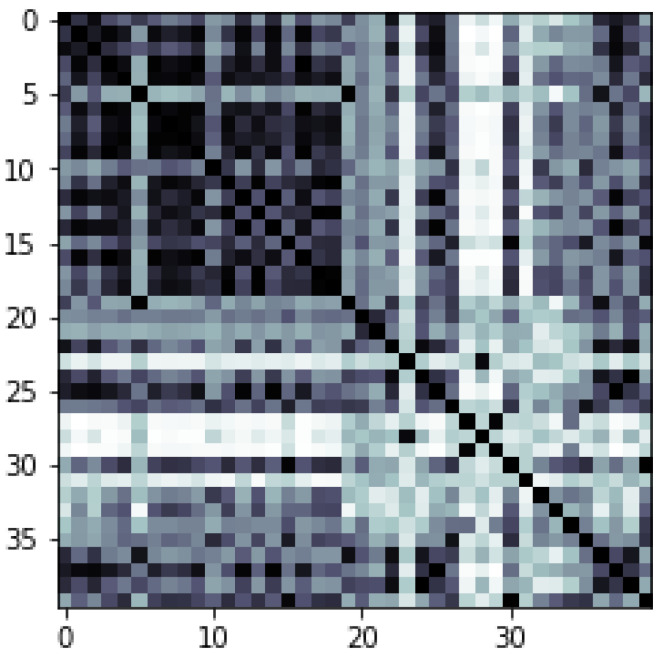
Kernel matrix of QSVM, with an accuracy of around 90%.

**Figure 17 entropy-25-00287-f017:**
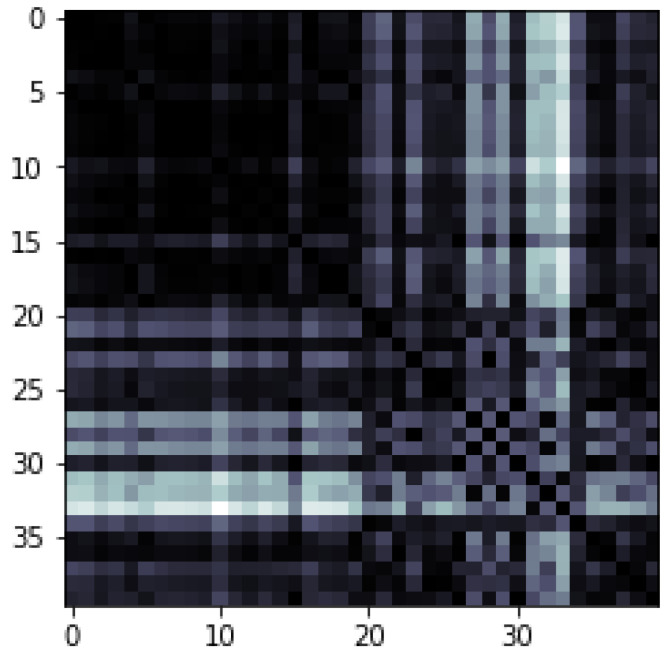
Kernel matrix of SVM, with an accuracy of around 85%.

**Figure 20 entropy-25-00287-f020:**
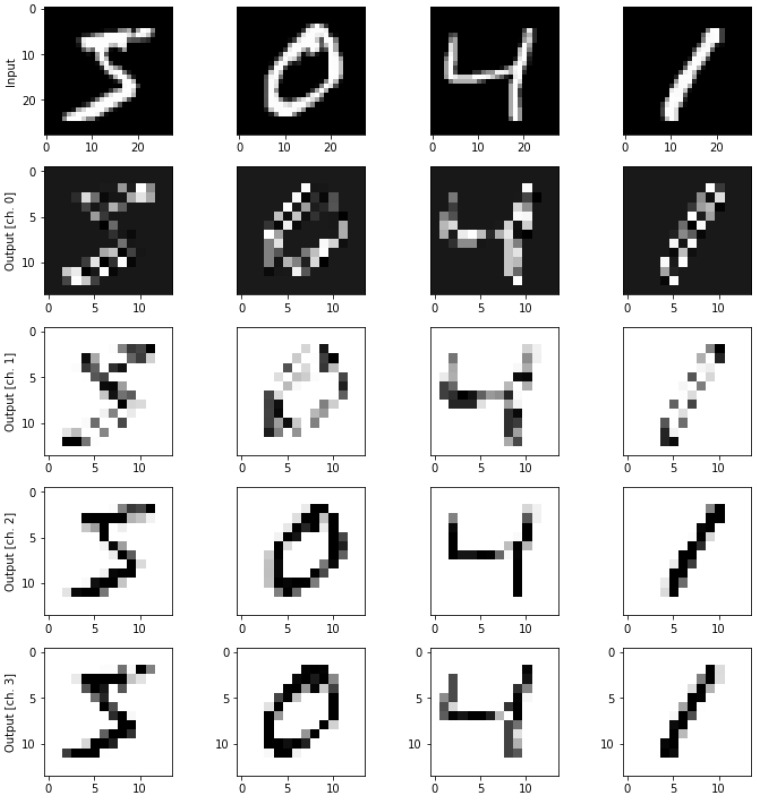
Under each input, the four outcome channels produced by our quantum convolution were shown in grayscale images.

**Figure 21 entropy-25-00287-f021:**
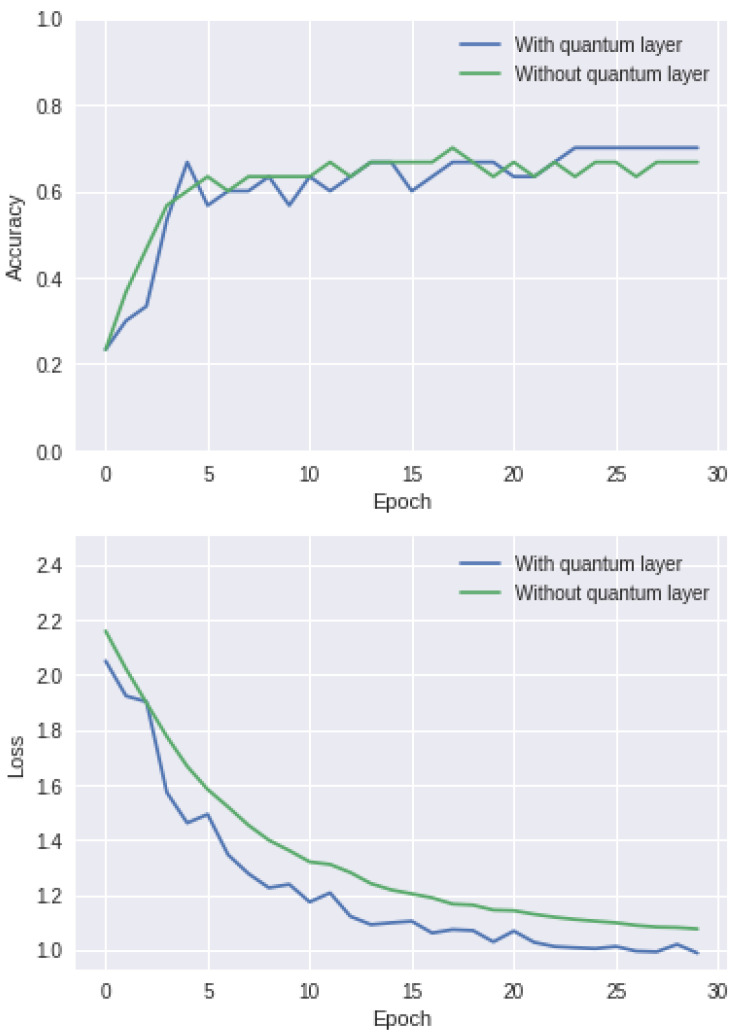
Comparative performance of the QCNN model and the CNN model.

**Table 1 entropy-25-00287-t001:** CNOT gate truth table.

Before Applying CNOT		After Applying CNOT	
**Controlled**	**Targeted**	**Controlled**	**Targeted**
|0〉	|0〉	|0〉	|0〉
|0〉	|1〉	|0〉	|1〉
|1〉	|0〉	|1〉	|1〉
|1〉	|1〉	|1〉	|0〉

**Table 2 entropy-25-00287-t002:** The results of the Bell states circuit computation.

Input	H	CNOT
|00〉	$\frac{|00\rangle +|10\rangle }{\sqrt{2}}$	$|\psi_{00}\rangle =\frac{|00\rangle +|11\rangle }{\sqrt{2}}$
|01〉	$\frac{|01\rangle +|11\rangle }{\sqrt{2}}$	$|\psi_{01}\rangle =\frac{|01\rangle +|10\rangle }{\sqrt{2}}$
|10〉	$\frac{|00\rangle -|10\rangle }{\sqrt{2}}$	$|\psi_{10}\rangle =\frac{|00\rangle -|11\rangle }{\sqrt{2}}$
|11〉	$\frac{|01\rangle -|11\rangle }{\sqrt{2}}$	$|\psi_{11}\rangle =\frac{|01\rangle -|10\rangle }{\sqrt{2}}$

**Table 3 entropy-25-00287-t003:** An overview of current quantum machine learning algorithm.

Quantum Routines	QML Applications
HHL algorithm	QSVM [4,74]
	Q linear regression [8]
	Q least squares [75]
	QPCA [14]
Grover’s algorithm	Q k-Means [10]
	Q K-Median [13]
	QKNN [6]
	Q Perceptron Models [76]
	Q Neural Networks [3]
Quantum phase estimation	Q k-Means [10]
Variational quantum circuit	Q decision tree [9]
	Circuit-centric quantum
	classifiers [77]
	Deep reinforcement
	learning [78]

**Table 4 entropy-25-00287-t004:** Result of the classification algorithms for the Iris dataset.

Metric	VQC	SVC	Decision Tree	Naive Bayes
Accuracy	1	0.96	0.96	0.96
Cost	0.2351166	0.0873071	0.0906691	1.381551

**Table 5 entropy-25-00287-t005:** Results of the benchmarking experiments.

Algorithm	Loss Function	Test Accuracy
**CNN**	1.0757	0.6667
**QNN**	0.9882	0.7000

## Data Availability

The notebooks used during this study can be provided after contacting the authors.

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
