# Peer review of "Quantum Machine Learning: A Review and Case Studies"

_entropy, 2023, doi:10.3390/e25020287_

Round 1
Reviewer 1 Report
In this work the authors review quantum machine learning in perspective of several conventional techniques in a point of view of computer scientist. The author discuss basic algorithms and implement Quanvolutional Neural Networks, Convolutional Neural Networks, QVSM, Variational Quantum Classifier and so on to solve some typical problems.
Quantum machine learning is a very hot and important topic in quantum computing.
This review is interesting and meaningful. It is well written and very comfortable to read. It is very nice review for beginner of quantum machine learning.
There are, however, a few issues that, while they appear readily fixable without affecting the results, might need to be fixed.
1. Line 124, there should be comma or full stop in the end of an equation. Similar problems in most of equations in this review.
2. Line 141 to line 143 , represent the equations. An equation can not be a line without connection with the lines.
For example,
“.... or a superposition of both,
c_1|0>+c_2|1>, (6)
where the coefficients c_1, c_2... ”
Make similar changes in other places.
3. There are some important papers on quantum machine learning which are worth mentioning in this review.
[1] Kishor Bharti, Alba Cervera-Lierta, Thi Ha Kyaw, Tobias Haug, Sumner Alperin-Lea, Abhinav Anand, Matthias Degroote, Hermanni Heimonen, Jakob S. Kottmann, Tim Menke, Wai-Keong Mok, Sukin Sim, Leong-Chuan Kwek, and Alán Aspuru-Guzik, Noisy intermediate-scale quantum algorithms. Reviews of Modern Physics, 2022, 94(1): 015004.
[2] H. Situ, Z. He, Y. Wang, L. Li, S. Zheng, Quantum generative adversarial network for generating discrete distribution, Information Sciences 538, 193-208 (2020). Also arXiv:1807.01235.
None of the above issues are major or appear to affect the review. I would be happy to recommend this paper to be accepted.
Author Response
Dear Reviewer,
We thank you and the other reviewers for their efforts in examining our work and offering insightful feedback. Your informative and important feedback led to potential enhancements to the current version. The authors have carefully reviewed the comments and made every effort to respond to each one. We expect that the revised version meets your requirements. The authors appreciate any more constructive feedback.
The replies are provided point-by-point below. All changes to the text are marked up using the “Track Changes” function in LaTeX(Overleaf).
[General Comment] In this work the authors review quantum machine learning in perspective of several conventional techniques in a point of view of computer scientist. The author discuss basic algorithms and implement Quanvolutional Neural Networks, Convolutional Neural Networks, QVSM, Variational Quantum Classifier and so on to solve some typical problems.
Quantum machine learning is a very hot and important topic in quantum computing.
This review is interesting and meaningful. It is well written and very comfortable to read. It is very nice review for beginner of quantum machine learning. There are, however, a few issues that, while they appear readily fixable without affecting the results, might need to be fixed. None of these issues are major or appear to affect the review. I would be happy to recommend this paper to be accepted.
Response: Thank you very much for agreeing with us on the purpose of this work. We have carefully reviewed your comments and tried our best to respond to each one individually.
[Minor comment 1] Line 124, there should be comma or full stop in the end of an equation. Similar problems in most of equations in this review.
Response: Thank you for your nice reminder. We went through the entire manuscript to fix this issue.
[Minor comment 2] Line 141 to line 143, represent the equations. An equation can not be a line without connection with the lines. For example:
“.... or a superposition of both,
c_1|0>+c_2|1>, (6)
where the coefficients c_1, c_2... ”
Make similar changes in other places.
Response: Thank you very much for the comment. Revised accordingly.
[Minor comment 3] There are some important papers on quantum machine learning which are worth mentioning in this review.
[1] Kishor Bharti, Alba Cervera-Lierta, Thi Ha Kyaw, Tobias Haug, Sumner Alperin-Lea, Abhinav Anand, Matthias Degroote, Hermanni Heimonen, Jakob S. Kottmann, Tim Menke, Wai-Keong Mok, Sukin Sim, Leong-Chuan Kwek, and Alán Aspuru-Guzik, Noisy intermediate-scale quantum algorithms. Reviews of Modern Physics, 2022, 94(1): 015004.
[2] H. Situ, Z. He, Y. Wang, L. Li, S. Zheng, Quantum generative adversarial network for generating discrete distribution, Information Sciences 538, 193-208 (2020). Also arXiv:1807.01235.
Response: We have cited the second reference that we found really worth mentioning in our review. Thank you.
Reviewer 2 Report
The authors have reviewed quantum machine learning based on near-term quantum computers, a recently focused topic in the field of quantum information processing. Specifically, the presented manuscript has shown the usefulness of typical quantum tasks such as the HHL Algorithm, Quantum Phase Estimation, and Variational Quantum Circuit, describing how to encode and utilize information into quantum states. In addition, it is practical in content since the authors have demonstrated the application for several tasks, introducing Qiskit and Pennylane software. Overall, this manuscript is written with nice readability, and the technical content is understandable. Thus, I recommend the publication of this manuscript after considering optional (minor) changes.
1) Quantum computation for a deeper quantum circuit is more powerful than that for a small-depth circuit. I believe that the authors should mention briefly quantum error correction and fault-tolerant quantum computation that can realize a deeper quantum circuit for the sake of providing prospects in this area.
Author Response
Dear Reviewer,
We thank you and the other reviewers for their efforts in examining our work and offering insightful feedback. Your informative and important feedback led to potential enhancements to the current version. The authors have carefully reviewed the comments and made every effort to respond to each one. We expect that the revised version meets your requirements. The authors appreciate any more constructive feedback.
[General Comment] The authors have reviewed quantum machine learning based on near-term quantum computers, a recently focused topic in the field of quantum information processing. Specifically, the presented manuscript has shown the usefulness of typical quantum tasks such as the HHL Algorithm, Quantum Phase Estimation, and Variational Quantum Circuit, describing how to encode and utilize information into quantum states. In addition, it is practical in content since the authors have demonstrated the application for several tasks, introducing Qiskit and Pennylane software. Overall, this manuscript is written with nice readability, and the technical content is understandable. Thus, I recommend the publication of this manuscript after considering optional (minor) changes.
Response: Thank you very much for recommending this review for publication.
[Minor comment] Quantum computation for a deeper quantum circuit is more powerful than that for a small-depth circuit. I believe that the authors should mention briefly quantum error correction and fault-tolerant quantum computation that can realize a deeper quantum circuit for the sake of providing prospects in this area.
Response: Thank you very much for the comment. As mentioned in our review, this work is a research path from fundamental quantum theory to quantum machine learning algorithms from a computer scientist's point of view. In order to simplify concepts for the reader, we will not dig into the physical aspects of quantum computing, such as qubit errors caused by quantum effects such as decoherence or uncontrolled state perturbation.
Reviewer 3 Report
The review article ‘Quantum Machine Learning: A Review and Case Studies’ by Zeguendry et al discusses the recent developments in quantum machine learning, in particular configurations where classical data is processed with quantum algorithms on quantum computers. They review the basic working, various classes of algorithms, their implementation and also the challenges involved. The review is timely and all relevant topics are well-covered, as well as benchmarking against the classical counterparts for the example cases is provided. Hence I recommend it for publication with the following suggestions:- - Captions of some figures are not very informative, e.g., the captions of Figures 9, 10, and 11 appear to be incomplete.
- - There are some typos and grammatical errors in sentence formation at places, the authors should correct them.
Author Response
Dear Reviewer,
We thank you and the other reviewers for their efforts in examining our work and offering insightful feedback. Your informative and important feedback led to potential enhancements to the current version. The authors have carefully reviewed the comments and made every effort to respond to each one. We expect that the revised version meets your requirements. The authors appreciate any more constructive feedback.
The replies are provided point-by-point below. All changes to the text are marked up using the “Track Changes” function in LaTeX(Overleaf).
[General Comment] The review article ‘Quantum Machine Learning: A Review and Case Studies’ by Zeguendry et al discusses the recent developments in quantum machine learning, in particular configurations where classical data is processed with quantum algorithms on quantum computers. They review the basic working, various classes of algorithms, their implementation and also the challenges involved. The review is timely and all relevant topics are well-covered, as well as benchmarking against the classical counterparts for the example cases is provided. Hence I recommend it for publication with the following suggestions.
Response: Thank you very much for recommending this review for publication.
[Minor comment 1] Captions of some figures are not very informative, e.g., the captions of Figures 9, 10, and 11 appear to be incomplete.
Response: Thank you for the nice reminder. Revised accordingly.
[Minor comment 2] There are some typos and grammatical errors in sentence formation at places, the authors should correct them.
Response: We went through the entire manuscript to eliminate grammatical mistakes. Thank you.